# Nonparametric Density Estimation with Adversarial Losses

**Shashank Singh**[1,2,*]     **Ananya Uppal**[3]     **Boyue Li**[4]
**Chun-Liang Li**[1]     **Manzil Zaheer**[1]     **Barnabás Póczos**[1]
[1]Machine Learning Department     [2]Department of Statistics & Data Science
[3]Department of Mathematical Sciences     [4]Language Technologies Institute
Carnegie Mellon University
[*]Corresponding Author: `sss1@cs.cmu.edu`

## Abstract

We study minimax convergence rates of nonparametric density estimation under a large class of loss functions called "adversarial losses", which, besides classical $\mathcal{L}^p$ losses, includes maximum mean discrepancy (MMD), Wasserstein distance, and total variation distance. These losses are closely related to the losses encoded by discriminator networks in generative adversarial networks (GANs). In a general framework, we study how the choice of loss and the assumed smoothness of the underlying density together determine the minimax rate. We also discuss implications for training GANs based on deep ReLU networks, and more general connections to learning implicit generative models in a minimax statistical sense.

## 1   Introduction

Generative modeling, that is, modeling the distribution from which data are drawn, is a central task in machine learning and statistics. Often, prior information is insufficient to guess the form of the data distribution. In statistics, generative modeling in these settings is usually studied from the perspective of nonparametric density estimation, in which histogram, kernel, orthogonal series, and nearest-neighbor methods are popular approaches with well-understood statistical properties [49, 47, 14, 7].

Recently, machine learning has made significant empirical progress in generative modeling, using such tools as generative adversarial networks (GANs) and variational autoencoders (VAEs). Computationally, these methods are quite distinct from classical density estimators; they usually rely on deep neural networks, fit by black-box optimization, rather than a mathematically prescribed smoothing operator, such as convolution with a kernel or projection onto a finite-dimensional subspace.

Ignoring the implementation of these models, from the perspective of statistical analysis, these recent methods have at least two main differences from classical density estimators. First, they are *implicit*, rather than *explicit* (or *prescriptive*) generative models [12, 29]; that is, rather than an estimate of the probability of a set or the density at a point, they return novel samples from the data distribution. Second, in many recent models, loss is measured not with $\mathcal{L}^p$ distances (as is conventional in nonparametric statistics [49, 47]), but rather with weaker losses, such as

$$d_{\mathcal{F}_D}(P, Q) = \sup_{f \in \mathcal{F}_D} \left| \mathbb{E}_{X \sim P}[f(X)] - \mathbb{E}_{X \sim Q}[f(X)] \right|, \tag{1}$$

where $\mathcal{F}_D$ is a *discriminator class* of bounded, Borel-measurable functions, and $P$ and $Q$ lie in a *generator class* $\mathcal{F}_G$ of Borel probability measures on a sample space $\mathcal{X}$. Specifically, GANs often use losses of this form because (1) can be approximated by a discriminator neural network.

This paper attempts to help bridge the gap between traditional nonparametric statistics and these recent advances by studying these two differences from a statistical minimax perspective. Specifically,

under traditional statistical smoothness assumptions, we identify (i.e., prove matching upper and lower bounds on) minimax convergence rates for density estimation under several losses of the form (1). We also discuss some consequences this has for particular neural network implementations of GANs based on these losses. Finally, we study connections between minimax rates for explicit and implicit generative modeling, under a plausible notion of risk for implicit generative models.

## 1.1 Adversarial Losses

The quantity (1) has been extensively studied, in the case that $\mathcal{F}_D$ is a reproducing kernel Hilbert space (RKHS) under the name *maximum mean discrepancy* (MMD; [18, 46]), and, in a wider context under the name *integral probability metric* (IPM; [31, 42, 43, 8]). [5] also called (1) the $\mathcal{F}_D$-*distance*, or, when $\mathcal{F}_D$ is a family of functions that can be implemented by a neural network, the *neural network distance*. We settled on the name "adversarial loss" because, without assuming any structure on $\mathcal{F}_D$, this matches the intuition of the expression (1), namely that of an adversary selecting the most distinguishing linear projection $f \in \mathcal{F}_D$ between the true density $P$ and our estimate $\widehat{P}$ (e.g., by the discriminator network in a GAN).

One can check that $d_{\mathcal{F}_D} : \mathcal{F}_G \times \mathcal{F}_G \to [0, \infty]$ is a pseudometric (i.e., it is non-negative and satisfies the triangle inequality, and $d_{\mathcal{F}_D}(P, Q) > 0 \Rightarrow P \neq Q$, although $d_{\mathcal{F}_D}(P, Q) = 0 \not\Rightarrow P = Q$ unless $\mathcal{F}_D$ is sufficiently rich). Many popular (pseudo)metrics between probability distributions, including $\mathcal{L}^p$ [49, 47], Sobolev [24, 30], maximum mean discrepancy (MMD; [46])/energy [45, 36], total variation [48], (1-)Wasserstein/Kantorovich-Rubinstein [20, 48], Kolmogorov-Smirnov [22, 40], and Dudley [13, 1] metrics can be written in this form, for appropriate choices of $\mathcal{F}_D$.

The **main contribution of this paper** is a statistical analysis of the problem of estimating a distribution $P$ from $n$ IID observations using the loss $d_{\mathcal{F}_D}$, in a minimax sense over $P \in \mathcal{F}_G$, for fairly general nonparametric smoothness classes $\mathcal{F}_D$ and $\mathcal{F}_G$. General upper and lower bounds are given in terms of decay rates of coefficients of functions in terms of an (arbitrary) orthonormal basis of $\mathcal{L}^2$ (including, e.g., Fourier or wavelet bases); note that this does *not* require $\mathcal{F}_D$ or $\mathcal{F}_G$ to have any inner product structure, only that $\mathcal{F}_D \subseteq \mathcal{L}^1$. We also discuss some consequences for density estimators based on neural networks (such as GANs), and consequences for the closely related problem of implicit generative modeling (i.e., of generating novel samples from a target distribution, rather than estimating the distribution itself), in terms of which GANs and VAEs are usually cast.

**Paper Organization:** Section 2 provides our formal problem statement and required notation. Section 3 discusses related work on nonparametric density estimation, with further discussion of the theory of GANs provided in the Appendix. Sections 4 and 5 contain our main theoretical upper and lower bound results, respectively. Section 6 develops our general results from Sections 4 and 5 into concrete minimax convergence rates for some important special cases. Section 7 uses our theoretical results to upper bound the error of perfectly optimized GANs. Section 8 establishes some theoretical relationships between the convergence of optimal density estimators and optimal implicit generative models. The Appendix provides proofs of our theoretical results, further applications, further discussion of related and future work, and experiments on simulated data that support our theoretical results.

## 2 Problem Statement and Notation

We now provide a formal statement of the problem studied in this paper in a very general setting, and then define notation required for our specific results.

**Formal Problem Statement:** Let $P \in \mathcal{F}_G$ be an unknown probability measure on a sample space $\mathcal{X}$, from which we observe $n$ IID samples $X_{1:n} = X_1, ..., X_n \overset{IID}{\sim} P$. In this paper, we are interested in using the samples $X_{1:n}$ to estimate the measure $P$, with error measured using the adversarial loss $d_{\mathcal{F}_D}$. Specifically, for various choices of spaces $\mathcal{F}_D$ and $\mathcal{F}_G$, we seek to bound the minimax rate

$$M(\mathcal{F}_D, \mathcal{F}_G) := \inf_{\widehat{P}} \sup_{P \in \mathcal{F}_G} \mathbb{E}_{X_{1:n}} \left[ d_{\mathcal{F}_D} \left( P, \widehat{P}(X_{1:n}) \right) \right]$$

of estimating distributions assumed to lie in a class $\mathcal{F}_G$, where the infimum is taken over all estimators $\widehat{P}$ (i.e., all (potentially randomized) functions $\widehat{P} : \mathcal{X}^n \to \mathcal{F}_G$). We will discuss both the case when $\mathcal{F}_G$ is known *a priori* and the *adaptive* case when it is not.

## 2.1 Notation

For a non-negative integer $n$, we use $[n] := \{1, 2, ..., n\}$ to denote the set of positive integers at most $n$. For sequences $\{a_n\}_{n \in \mathbb{N}}$ and $\{b_n\}_{n \in \mathbb{N}}$ of non-negative reals, $a_n \lesssim b_n$ and, similarly $b_n \gtrsim a_n$, indicate the existence of a constant $C > 0$ such that $\limsup_{n \to \infty} \frac{a_n}{b_n} \leq C$. $a_n \asymp b_n$ indicates $a_n \lesssim b_n \lesssim a_n$. For functions $f : \mathbb{R}^d \to \mathbb{R}$, we write

$$\lim_{\|z\| \to \infty} f(z) := \sup_{\{z_n\}_{n \in \mathbb{N}} : \|z_n\| \to \infty} \lim_{n \to \infty} f(z_n),$$

where the supremum is taken over all diverging $\mathbb{R}^d$-valued sequences. Note that, by equivalence of finite-dimensional norms, the exact choice of the norm $\| \cdot \|$ does not matter here. We will also require summations of the form $\sum_{z \in \mathcal{Z}} f(z)$ in cases where $\mathcal{Z}$ is a (potentially infinite) countable index set and $\{f(z)\}_{z \in \mathcal{Z}}$ is summable but not necessarily absolutely summable. Therefore, to ensure that the summation is well-defined, the order of summation will need to be specified, depending on the application (as in, e.g., Section 6).

Fix the sample space $\mathcal{X} = [0, 1]^d$ to be the $d$-dimensional unit cube, over which $\lambda$ denotes the usual Lebesgue measure. Given a measurable function $f : \mathcal{X} \to \mathbb{R}$, let, for any Borel measure $\mu$ on $\mathcal{X}$, $p \in [1, \infty]$, and $L > 0$,

$$\|f\|_{\mathcal{L}_\mu^p} := \left( \int_{\mathcal{X}} |f|^p \, d\mu \right)^{1/p} \quad \text{and} \quad \mathcal{L}_\mu^p(L) := \left\{ f : \mathcal{X} \to \mathbb{R} \,\middle|\, \|f\|_{\mathcal{L}_\mu^p} < L \right\}$$

(taking the appropriate limit if $p = \infty$) denote the Lebesgue norm and ball of radius $L$, respectively.

Fix an orthonormal basis $\mathcal{B} = \{\phi_z\}_{z \in \mathcal{Z}}$ of $\mathcal{L}_\lambda^2$ indexed by a countable family $\mathcal{Z}$. To allow probability measures $P$ without densities (i.e., $P \not\ll \mu$), we assume each basis element $\phi_z : \mathcal{X} \to \mathbb{R}$ is a bounded function, so that $\widetilde{P}_z := \mathbb{E}_{X \sim P} [\phi_z(X)]$ is well-defined. For constants $L > 0$ and $p \geq 1$ and real-valued net $\{a_z\}_{z \in \mathcal{Z}}$, our results pertain to generalized ellipses of the form

$$\mathcal{H}_{p,a}(L) = \left\{ f \in \mathcal{L}^1(\mathcal{X}) : \left( \sum_{z \in \mathcal{Z}} a_z^p |\widetilde{f}_z|^p \right)^{1/p} \leq L \right\}.$$

(where $\widetilde{f}_z := \int_{\mathcal{X}} f \phi_z \, d\mu$ is the $z^{th}$ coefficient of $f$ in the basis $\mathcal{B}$). We sometimes omit dependence on $L$ (e.g., $\mathcal{H}_{p,a} = \mathcal{H}_{p,a}(L)$) when its value does not matter (e.g., when discussing *rates* of convergence).

A particular case of interest is the scale of the Sobolev spaces defined for $s, L \geq 0$ and $p \geq 1$ by

$$\mathcal{W}^{s,p}(L) = \left\{ f \in \mathcal{L}^1(\mathcal{X}) : \left( \sum_{z \in \mathcal{Z}} |z|^{sp} |\widetilde{f}_z|^p \right)^{1/p} \leq L \right\}.$$

For example, when $\mathcal{B}$ is the standard Fourier basis and $s$ is an integer, for a constant factor $c$ depending only on $s$ and the dimension $d$,

$$\mathcal{W}^{s,p}(cL) := \left\{ f \in \mathcal{L}_\lambda^p \,\middle|\, \left\| f^{(s)} \right\|_{\mathcal{L}_\lambda^p} < L \right\}$$

corresponds to the natural standard smoothness class of $\mathcal{L}_\lambda^p$ functions having $s^{th}$-order (weak) derivatives $f^{(s)}$ in $\mathcal{L}_\lambda^p(L)$ [24]).

## 3 Related Work

Our results apply directly to many of the losses that have been used in GANs, including 1-Wasserstein distance [3, 19], MMD [25], Sobolev distances [30], and the Dudley metric [1]. As discussed in the Appendix, slightly different assumptions are required to obtain results for the Jensen-Shannon divergence (used in the original GAN formulation of [17]) and other $f$-divergences [33].

Given their generality, our results relate to many prior works on distribution estimation, including classical work in nonparametric statistics and empirical process theory, as well as more recent work

studying Wasserstein distances and MMD. Here, we briefly survey known results for these problems. There have also been a few other statistical analyses of the GAN framework; due to space constraints, we discuss these works in the Appendix.

$\mathcal{L}_\lambda^2$ **distances:** Classical work on nonparametric statistics has typically focused on the problem of smooth density estimation under $\mathcal{L}_\lambda^2$ loss, corresponding the adversarial loss $d_{\mathcal{F}_D}$ with $\mathcal{F}_D = \mathcal{L}_\lambda^2(L_D)$ (the Hölder dual) of $\mathcal{L}^2$ [49, 47]. In this case, when $\mathcal{F}_G = \mathcal{W}^{t,2}(L_G)$ is a Sobolev class, then the minimax rate is typically $M(\mathcal{F}_D, \mathcal{F}_G) \asymp n^{-\frac{t}{2t+d}}$, matching the rates given by our main results.

**Maximum Mean Discrepancy (MMD):** When $\mathcal{F}_D$ is a reproducing kernel Hilbert space (RKHS), the adversarial loss $d_{\mathcal{F}_D}$ has been widely studied under the name *maximum mean discrepancy (MMD)* [18, 46]. When the RKHS kernel is translation-invariant, one can express $\mathcal{F}_D$ in the form $\mathcal{H}_{2,a}$, where $a$ is determined by the spectrum of the kernel, and so our analysis holds for MMD losses with translation-invariant kernels (see Example 6). To the best of our knowledge, minimax rates for density estimation under MMD loss have not been established in general; our analysis suggests that density estimation under an MMD loss is essentially equivalent to the problem of estimating kernel mean embeddings studied in [46], as both amount to density estimation while ignoring bias, and both typically have a parametric $n^{-1/2}$ minimax rate. Note that the related problems of estimating MMD itself, and of using it in statistical tests for homogeneity and dependence, have received extensive theoretical treatment [18, 35].

**Wasserstein Distances:** When $\mathcal{F}_D = \mathcal{W}^{1,\infty}(L)$ is the class of 1-Lipschitz functions, $d_{\mathcal{F}_D}$ is equivalent to the *(order-1) Wasserstein* (also called *earth-mover's* or *Kantorovich-Rubinstein*) distance. In this case, when $\mathcal{F}_G$ contains all Borel measurable distributions on $\mathcal{X}$, minimax bounds have been established under very general conditions (essentially, when the sample space $\mathcal{X}$ is an arbitrary totally bounded metric space) in terms of covering numbers of $\mathcal{X}$ [50, 39, 23]. In the particular case that $\mathcal{X}$ is a bounded subset of $\mathbb{R}^d$ of full dimension (i.e., having non-empty interior, comparable to the case $\mathcal{X} = [0,1]^d$ that we study here), these results imply a minimax rate of $M(\mathcal{F}_D, \mathcal{F}_G) = n^{-\min\{\frac{1}{2}, \frac{1}{d}\}}$, matching our rates. Notably, these upper bounds are derived using the empirical distribution, which *cannot* benefit from smoothness of the true distribution (see [50]). At the same time, it is obvious to generalize smoothing estimators to sample spaces that are not sufficiently nice subsets of $\mathbb{R}^d$.

**Sobolev IPMs:** The closest work to the present is [26], which we believe was the first work to analyze how convergence rates jointly depend on (Sobolev) smoothness restrictions on both $\mathcal{F}_D$ and $\mathcal{F}_G$. Specifically, for Sobolev spaces $\mathcal{F}_D = \mathcal{W}^{s,p}$ and $\mathcal{F}_G = \mathcal{W}^{t,q}$ with $p, q \geq 2$ (compare our Example 4), they showed

$$n^{-\frac{s+t}{2t+d}} \lesssim M(\mathcal{W}^{s,2}, \mathcal{W}^{t,2}) \lesssim n^{-\frac{s+t}{2(s+t)+d}}. \tag{2}$$

Our main results in Sections 4 and 5 improve on this in two main ways. First, our results generalize to and are tight for many spaces besides Sobolev spaces. Examples include when $\mathcal{F}_D$ is a reproducing kernel Hilbert space (RKHS) with translation-invariant kernel, or when $\mathcal{F}_G$ is the class of all Borel probability measures. Our bounds also allow other (e.g., wavelet) estimators, whereas the bounds of [26] are for the (uniformly $\mathcal{L}_\lambda^\infty$-bounded) Fourier basis. Second, the lower and upper bounds in (2) diverge by a factor polynomial in $n$. We tighten the upper bound to match the lower bound, identifying, for the first time, minimax rates for many problems of this form (e.g., $M(\mathcal{W}^{s,2}, \mathcal{W}^{t,2}) \asymp n^{-\frac{s+t}{2t+d}}$ in the Sobolev case above). Our analysis has several interesting implications:

1. When $s > d/2$, the convergence becomes *parametric*: $M(W^{s,2}, \mathcal{F}_G) \asymp n^{-1/2}$, for *any class of distributions* $\mathcal{F}_G$. This highlights that the loss $d_{\mathcal{F}_D}$ is quite weak for large $s$, and matches known minimax results for the Wasserstein case $s = 1$ [10, 39].

2. Our upper bounds, as in [26], are for smoothing estimators (namely, the orthogonal series estimator 3). In contrast, previous analyses of Wasserstein loss focused on convergence of the (unsmoothed) empirical distribution $\widehat{P}_E$ to the true distribution, which typically occurs at rate of $\asymp n^{-1/d} + n^{-1/2}$, where $d$ is the intrinsic dimension of the support of $P$ [10, 50, 39]. Moreover, if $\mathcal{F}_G$ includes all Borel probability measures, this rate is minimax optimal [39]. The loose upper bound of [26] left open the questions of whether (when $s < d/2$) a very small amount ($t \in \left(0, \frac{2s^2}{d-2s}\right]$) of smoothness improves the minimax rate and, more importantly, whether smoothed estimators are outperformed by $\widehat{P}_E$ in this regime. Our results imply that, for $s < d/2$, the minimax rate strictly improves with smoothness $t$, and that, as long as the support of $P$

has full dimension, the smoothed estimator *always* converges faster than $\widehat{P}_E$. An important open problem is to simultaneously leverage when $P$ is smooth *and* has support of low intrinsic dimension; many data (e.g., images) likely enjoy both these properties.

3. [26] suggested over-smoothing the estimate (the smoothing parameter $\zeta$ discussed in Equation (3) below was set to $\zeta \asymp n^{\frac{1}{2(s+t)+d}}$) compared to the case of $\mathcal{L}_\lambda^2$ loss, and hence it was not clear how to design estimators that adapt to unknown smoothness under losses $d_{W^{s,p}}$. We show that the optimal smoothing ($\zeta \asymp n^{\frac{1}{2t+d}}$) under $d_{W^{s,p}}$ loss is identical to that under $\mathcal{L}_\lambda^2$ loss, and we use this to design an adaptive estimator (see Corollary 5).

4. Our bounds imply improved performance bounds for optimized GANs, discussed in Section 7.

## 4  Upper Bounds for Orthogonal Series Estimators

This section gives upper bounds on the adversarial risk of the following density estimator. For any finite set $Z \subseteq \mathcal{Z}$, let $\widehat{P}_Z$ be the truncated series estimate

$$\widehat{P}_Z := \sum_{z \in Z} \widehat{P}_z \phi_z, \quad \text{where, for any } z \in \mathcal{Z}, \quad \widehat{P}_z := \frac{1}{n} \sum_{i=1}^{n} \phi_z(X_i). \tag{3}$$

$Z$ is a tuning parameter that typically corresponds to a smoothing parameter; for example, when $\mathcal{B}$ is the Fourier basis and $Z = \{z \in \mathbb{Z}^d : \|z\|_\infty \le \zeta\}$ for some $\zeta > 0$, $\widehat{P}_Z$ is equivalent to a kernel density estimator using a sinc product kernel $K_h(x) = \prod_{j=1}^{d} \frac{2}{h} \frac{\sin(2\pi x/h)}{2\pi x/h}$ with bandwidth $h = 1/\zeta$ [34].

We now present our main upper bound on the minimax rate of density estimation under adversarial losses. The upper bound is given by the orthogonal series estimator given in Equation (3), but we expect kernel and other standard linear density estimators to converge at the same rate.

**Theorem 1** (Upper Bound). *Suppose that $\mu(\mathcal{X}) < \infty$ and there exist constants $L_D, L_G > 0$, real-valued nets $\{a_z\}_{z \in \mathcal{Z}}$, $\{b_z\}_{z \in \mathcal{Z}}$ such that $\mathcal{F}_D = \mathcal{H}_{p,a}(\mathcal{X}, L_D)$ and $\mathcal{F}_G = \mathcal{H}_{q,b}(\mathcal{X}, L_G)$, where $p, q \ge 1$. Let $p' = \frac{p}{p-1}$ denote the Hölder conjugate of $p$. Then, for any $P \in \mathcal{F}_G$,*

$$\underset{X_{1:n}}{\mathbb{E}} \left[ d_{\mathcal{F}_D} \left( P, \widehat{P} \right) \right] \le L_D \frac{c_{p'}}{\sqrt{n}} \left\| \left\{ \frac{\|\phi_z\|_{\mathcal{L}_P^\infty}}{a_z} \right\}_{z \in Z} \right\|_{p'} + L_D L_G \left\| \left\{ \frac{1}{a_z b_z} \right\}_{z \in \mathcal{Z} \setminus Z} \right\|_{\frac{1}{1-1/p-1/q}} \tag{4}$$

The two terms in the bound (4) demonstrate a bias-variance tradeoff, in which the first term (*variance*) increases with the truncation set $Z$ and is typically independent of the class $\mathcal{F}_G$ of distributions, while the second term (*bias*) decreases with $Z$ at a rate depending on the complexity of $\mathcal{F}_G$.

**Corollary 2** (Sufficient Conditions for Parametric Rate). *Consider the setting of Theorem 1. If*

$$A := \sum_{z \in \mathcal{Z}} \frac{\|\phi_z\|_{\mathcal{L}_P^\infty}^2}{a_z^2} < \infty \quad \text{and} \quad \max\{a_z, b_z\} \to \infty.$$

*whenever $\|z\| \to \infty$, then, the minimax rate is parametric; specifically, $M(\mathcal{F}_D, \mathcal{F}_G) \le L_D \sqrt{A/n}$. In particular, letting $c_z := \sup_{x \in \mathcal{X}} |\phi_z(x)|$ for each $z \in \mathcal{Z}$, this occurs whenever $\sum_{z \in \mathcal{Z}} \frac{c_z^2}{a_z^2} < \infty$.*

In many contexts (e.g., if $P \ll \lambda$ and $\lambda \ll P$), the simpler condition $\sum_{z \in \mathcal{Z}} \frac{c_z^2}{a_z^2} < \infty$ suffices. The first, and slightly weaker condition in terms of $\|\phi_z\|_{\mathcal{L}_P^\infty}^2$ is useful when we restrict $\mathcal{F}_G$; e.g., if $\mathcal{B}$ is the wavelet basis (defined in the Appendix) and $\mathcal{F}_G$ contains only discrete distributions supported on at most $k$ points, then $\|\phi_{i,j}\|_{\mathcal{L}_P^\infty}^2 = 0$ for all but $k$ values of $j \in [2^i]$, at each resolution $i \in \mathbb{N}$. The assumption $\max\left\{\lim_{\|z\|\to\infty} a_z, \lim_{\|z\|\to\infty} b_z\right\} = \infty$ is quite mild; for example, the Riemann-Lebesgue lemma and the assumption that $\mathcal{F}_D$ is bounded in $\mathcal{L}_\lambda^\infty \subseteq \mathcal{L}_\lambda^1$ together imply that this condition always holds if $\mathcal{B}$ is the Fourier basis.

## 5  Minimax Lower Bound

In this section, we lower bound the minimax risk $M(\mathcal{F}_D, \mathcal{F}_G)$ of distribution estimation under $d_{\mathcal{F}_D}$ loss over $\mathcal{F}_G$, for the case when $\mathcal{F}_D = \mathcal{H}_{p,a}$ and $\mathcal{F}_G := \mathcal{H}_{q,b}$ are generalized ellipses. As we show

in some examples in Section 6, our lower bound rate matches our upper bound rate in Theorem 1 for many spaces $\mathcal{F}_D$ and $\mathcal{F}_G$ of interest. Our lower bound also suggests that the assumptions in Corollary 2 are typically necessary to guarantee the parametric convergence rate $n^{-1/2}$.

**Theorem 3** (Minimax Lower Bound). *Fix $\mathcal{X} = [0, 1]^d$, and let $p_0$ denote the uniform density (with respect to Lebesgue measure) on $\mathcal{X}$. Suppose $\{p_0\} \cup \{\phi_z\}_{z \in \mathcal{Z}}$ is an orthonormal basis in $\mathcal{L}_\mu^2$, and $\{a_z\}_{z \in \mathcal{Z}}$ and $\{b_z\}_{z \in \mathcal{Z}}$ are two real-valued nets. Let $L_D, L_G \geq 0$ and $p, q \geq 2$. For any $Z \subseteq \mathcal{Z}$, let*

$$A_Z := |Z|^{1/2} \sup_{z \in Z} a_z \quad and \quad B_Z := |Z|^{1/2} \sup_{z \in Z} b_z.$$

*Then, for $\mathcal{F}_D = \mathcal{H}_{p,a}(L_D)$ and $\mathcal{F}_G := \mathcal{H}_{q,b}(L_G)$, for any $Z \subseteq \mathcal{Z}$ satisfying*

$$B_Z \geq 16 L_G \sqrt{\frac{n}{\log 2}} \quad and \quad 2 \frac{L_G}{B_Z} \sum_{z \in Z} \|\phi_z\|_{\mathcal{L}_\mu^\infty} \leq 1, \tag{5}$$

*we have $M(\mathcal{F}_D, \mathcal{F}_G) \geq \dfrac{L_G L_D |Z|}{64 A_Z B_Z} = \dfrac{L_G L_D}{64 \left(\sup_{z \in Z} a_z\right)\left(\sup_{z \in Z} b_z\right)}.$*

As in most minimax lower bounds, our proof relies on constructing a finite set $\Omega_G$ of "worst-case" densities in $\mathcal{F}_G$, lower bounding the distance $d_{\mathcal{F}_D}$ over $\Omega_G$, and then letting elements of $\Omega_G$ shrink towards the uniform distribution $p_0$ at a rate such that the average information (here, Kullback-Leibler) divergence between each $p \in \Omega_G$ and $p_0$ does not grow with $n$. The first condition in (5) ensures that the information divergence between each $p \in \Omega_G$ and $p_0$ is sufficiently small, and typically results in tuning of $Z$ identical (in rate) to its optimal tuning in the upper bound (Theorem 1).

The second condition in (5) is needed to ensure that the "worst-case" densities we construct are everywhere non-negative. Hence, this condition is not needed for lower bounds in the Gaussian sequence model, as in Theorem 2.3 of [26]. However, failure of this condition (asymptotically) corresponds to the breakdown point of the asymptotic equivalence between the Gaussian sequence model and the density estimation model in the regime of very low smoothness (e.g., in the Sobolev setting, when $t < d/2$; see [9]), and so finer analysis is needed to establish lower bounds here.

# 6 Examples

In this section, we apply our bounds from Sections 4 and 5 to compute concrete minimax convergence rates for two examples choices of $\mathcal{F}_D$ and $\mathcal{F}_G$, namely Sobolev spaces and reproducing kernel Hilbert spaces. Due to space constraints, we consider only the Fourier basis here, but, in the Appendix, we also discuss an estimator in the Sobolev case using the Haar wavelet basis.

For the purpose of this section, suppose that $\mathcal{X} = [0, 2\pi]^d$, $\mathcal{Z} = \mathbb{Z}^d$, and, for each $z \in \mathcal{Z}$, $\phi_z$ is the $z^{th}$ standard Fourier basis element given by $\phi_z(x) = e^{i\langle z, x \rangle}$ for all $x \in \mathcal{X}$. In this case, we will always choose the truncation set $Z$ to be of the form $Z := \{z \in \mathcal{Z} : \|z\|_\infty \leq \zeta\}$, for some $\zeta > 0$, so that $|Z| \leq \zeta^d$. Moreover, for every $z \in Z$, $\|\phi_z\|_{\mathcal{L}_\mu^\infty} = 1$, and hence $C_Z \leq 1$.

**Example 4** (Sobolev Spaces). Suppose that, for some $s, t \geq 0$, $a_z = \|z\|_\infty^s$ and $b_z = \|z\|_\infty^t$. Then, setting $\zeta = n^{\frac{1}{2t+d}}$ in Theorems 1 and 3 gives that there exist constants $C > c > 0$ such that

$$cn^{-\min\left\{\frac{1}{2}, \frac{s+t}{2t+d}\right\}} \leq M\left(\mathcal{W}^{s,2}, \mathcal{W}^{t,2}\right) \leq Cn^{-\min\left\{\frac{1}{2}, \frac{s+t}{2t+d}\right\}}. \tag{6}$$

Combining the observation that the $s$-Hölder space $\mathcal{W}^{s,\infty} \subseteq \mathcal{W}^{s,2}$ with the lower bound (over $\mathcal{W}^{s,\infty}$) in Theorem 3.1 of [26], we have that (6) also holds when $\mathcal{W}^{s,2}$ is replaced with $\mathcal{W}^{s,p}$ for any $p \in [2, \infty]$ (e.g., in the case of the Wasserstein metric $d_{\mathcal{W}^{1,\infty}}$).

So far, we have assumed the smoothness $t$ of the true distribution $P$ is known, and used that to tune the parameter $\zeta$ of the estimator. However, in reality, $t$ is not known. In the next result, we leverage the fact that the rate-optimal choice $\zeta = n^{\frac{1}{2t+d}}$ above does not rely on the loss parameters $s$, together with Theorem 1 to construct an *adaptively minimax estimator*, i.e., one that is minimax and fully-data dependent. There is a large literature on adaptive nonparametric density estimation under $\mathcal{L}_\mu^2$ loss; see [14] for accessible high-level discussion and [16] for a technical but comprehensive review.

**Corollary 5** (Adaptive Upper Bound for Sobolev Spaces). *There exists an adaptive choice $\widehat{\zeta} : \mathcal{X}^n \to \mathbb{N}$ of the hyperparameter $\zeta$ (independent of $s,t$), such that, for any $s,t \geq 0$, there exists a constant $C > 0$ (independent of $n$), such that*

$$\sup_{P \in \mathcal{W}^{t,2}} \mathbb{E}_{X_{1:n} \overset{IID}{\sim} P} \left[ d_{\mathcal{W}^{s,2}} \left( P, \widehat{P}_{Z_{\widehat{\zeta}(X_{1:n})}} \right) \right] \leq M \left( \mathcal{W}^{s,2}, \mathcal{W}^{t,2} \right) \tag{7}$$

Due to space constraints, we present the actual construction of the adaptive $\widehat{\zeta}$ in the Appendix, but, in brief, it is a standard construction based on leave-one-out cross-validation under $\mathcal{L}^2_\mu$ loss which is known (e.g., see Sections 7.2.1 and 7.5.2 of [28]) to be adaptively minimax under $\mathcal{L}^2_\mu$ loss. Using the fact that our upper bound Theorem 1 uses a choice of $\zeta$ is independent of the loss parameter $s$, we show that the $d_{\mathcal{W}^{s,\infty}}$ risk of $\widehat{P}_\zeta$ can be factored into its $\mathcal{L}^2_\mu$ risk and a component ($\zeta^{-s}$) that is independent of $t$. Since $\mathcal{L}^2_\mu$ risk can be rate-minimized in independently of $t$, it follows that the $d_{\mathcal{W}^{s,\infty}}$ risk can be rate-minimized independently of $t$. Adaptive minimaxity then follows from Theorem 3.

**Example 6** (Reproducing Kernel Hilbert Space/MMD Loss). Suppose $\mathcal{H}_k$ is a reproducing kernel Hilbert space (RKHS) with reproducing kernel $k : \mathcal{X} \times \mathcal{X} \to \mathbb{R}$ [4, 6]. If $k$ is translation invariant (i.e., there exists $\kappa \in \mathcal{L}^2_\mu$ such that, for all $x,y \in \mathcal{X}$, $k(x,y) = \kappa(x-y)$), then Bochner's theorem (see, e.g., Theorem 6.6 of [51]) implies that, up to constant factors,

$$\mathcal{H}_k(L) := \{ f \in \mathcal{H}_k : \|f\|_{\mathcal{H}_k} \leq L \} = \left\{ f \in \mathcal{H}_k : \sum_{z \in \mathcal{Z}} |\widetilde{\kappa}_z|^2 |\widetilde{f}_z|^2 < L^2 \right\}.$$

Thus, in the setting of Theorem 1, we have $\mathcal{H}_k = \mathcal{H}_{2,a}$, where $a_z = |\widetilde{\kappa}_z|$ satisfies $\sum_{z \in \mathcal{Z}} a_z^{-2} = \|\kappa\|^2_{\mathcal{L}^2_\mu} < \infty$. Corollary 2 then gives $M(\mathcal{H}_k(L_D), \mathcal{F}_G) \leq L_D \|\kappa\|_{\mathcal{L}^2_\mu} n^{-1/2}$ for *any class* $\mathcal{F}_G$. It is well-known known that MMD can always be *estimated* at the parametric rate $n^{-1/2}$ [18]; however, to the best of our knowledge, only recently has it been shown that any probability distribution can be estimated at the rate $n^{-1/2}$ under MMD loss[41], emphasizing the fact that MMD is a very weak metric. This has important implications for applications such as two-sample testing [35].

# 7 Consequences for Generative Adversarial Neural Networks (GANs)

This section discusses implications of our minimax bounds for GANs. Neural networks in this section are assumed to be fully-connected, with rectified linear unit (ReLU) activations. [26] used their upper bound result (2) to prove a similar theorem, but, since their upper bound was loose, the resulting theorem was also loose. The following results are immediate consequences of our improvement (Theorem 1) over the upper bound (2) of [26], and so we refer to that paper for the proof. Key ingredients are an oracle inequality proven in [26], an upper bound such as Theorem 1, and bounds of [52] on the size of a neural network needed to approximate functions in a Sobolev class.

In the following, $\mathcal{F}_D$ denotes the set of functions that can be encoded by the discriminator network and $\mathcal{F}_G$ denotes the set of distributions that can be encoded by the generator network. $P_n := \frac{1}{n} \sum_{i=1}^n 1_{\{X_i\}}$ denotes the empirical distribution of the observed data $X_{1:n} \overset{IID}{\sim} P$.

**Theorem 7** (Improvement of Theorem 3.1 in Liang [26]). *Let $s,t > 0$, and fix a desired approximation accuracy $\epsilon > 0$. Then, there exists a GAN architecture, in which*

1. *the discriminator $\mathcal{F}_D$ has at most $O(\log(1/\epsilon))$ layers and $O(\epsilon^{-d/s} \log(1/\epsilon))$ parameters,*
2. *and the generator $\mathcal{F}_G$ has at most $O(\log(1/\epsilon))$ layers and $O(\epsilon^{-d/t} \log(1/\epsilon))$ parameters,*

*such that, if $\widehat{P}_*(X_{1:n}) := \underset{\widehat{P} \in \mathcal{F}_G}{\operatorname{argmin}} d_{\mathcal{F}_D} \left( P_n, \widehat{P} \right)$, is the optimized GAN estimate of $P$,*

*then* $\displaystyle \sup_{P \in \mathcal{W}^{t,2}} \mathbb{E}_{X_{1:n}} \left[ d_{\mathcal{W}^{s,2}} \left( P, \widehat{P}_*(X_{1:n}) \right) \right] \leq C \left( \epsilon + n^{-\min\left\{ \frac{1}{2}, \frac{s+t}{2t+d} \right\}} \right).$

The discriminator and generator in the above theorem can be implemented as described in [52]. The assumption that the GAN is perfectly optimized may be strong; see [32, 27] for discussion of this.

Though we do not present this result due to space constraints, we can similarly improve the upper bound of [26] (their Theorem 3.2) for very deep neural networks, further improving on the previous state-of-the-art bounds of [2] (which did not leverage smoothness assumptions on $P$).

# 8 Minimax Comparison of Explicit and Implicit Generative Models

In this section, we draw formal connections between our work on density estimation (explicit generative modeling) and the problem of implicit generative modeling under an appropriate measure of risk. In the sequel, we fix a class $\mathcal{F}_G$ of probability measures on a sample space $\mathcal{X}$ and a loss function $\ell : \mathcal{F}_G \times \mathcal{F}_G \to [0, \infty]$ measuring the distance of an estimate $\widehat{P}$ from the true distribution $P$. $\ell$ need not be an adversarial loss $d_{\mathcal{F}_D}$, but our discussion does apply to all $\ell$ of this form.

## 8.1 A Minimax Framework for Implicit Generative Models

Thus far, we have analyzed the *minimax risk of density estimation*, namely

$$M_D(\mathcal{F}_G, \ell, n) = \inf_{\widehat{P}} \sup_{P \in \mathcal{F}_G} R_D(P, \widehat{P}), \text{ where } R_D(P, \widehat{P}) = \mathbb{E}_{X_{1:n} \overset{IID}{\sim} P} \left[ \ell(P, \widehat{P}(X_{1:n})) \right] \qquad (8)$$

denotes the *density estimation risk of* $\widehat{P}$ *at* $P$ and the infimum is taken over all estimators (i.e., (potentially randomized) functions $\widehat{P} : \mathcal{X}^n \to \mathcal{F}_G$). Whereas density estimation is a classical statistical problem to which we have already contributed novel results, our motivations for studying this problem arose from a desire to better understand recent work on implicit generative modeling.

Implicit generative models, such as GANs [3, 17] and VAEs [21, 37], address the problem of *sampling*, in which we seek to construct a *generator* that produces novel samples from the distribution $P$ [29]. In our context, a generator is a function $\widehat{X} : \mathcal{X}^n \times \mathcal{Z} \to \mathcal{X}$ that takes in $n$ IID samples $X_{1:n} \sim P$ and a source of randomness (a.k.a., *latent variable*) $Z \sim Q_Z$ with known distribution $Q_Z$ (independent of $X_{1:n}$) on a space $\mathcal{Z}$, and returns a novel sample $\widehat{X}(X_{1:n}, Z) \in \mathcal{X}$.

The evaluating the performance of implicit generative models, both in theory and in practice, is difficult, with solutions continuing to be proposed [44], some of which have proven controversial. Some of this controversy stems from the fact that many of the most straightforward evaluation objectives are optimized by a trivial generator that 'memorizes' the training data (e.g., $\widehat{X}(X_{1:n}, Z) = X_Z$, where $Z$ is uniformly distributed on $[n]$). One objective that can avoid this problem is as follows. For simplicity, fix the distribution $Q_Z$ of the latent random variable $Z \sim Q_Z$ (e.g., $Q_Z = \mathcal{N}(0, I)$). For a fixed training set $X_{1:n} \overset{IID}{\sim} P$ and latent distribution $Z \sim Q_Z$, we define the *implicit distribution of a generator* $\widehat{X}$ as the conditional distribution $P_{\widehat{X}(X_{1:n}, Z)|X_{1:n}}$ over $\mathcal{X}$ of the random variable $\widehat{X}(X_{1:n}, Z)$ given the training data. Then, for any $P \in \mathcal{F}_G$, we define the *implicit risk of* $\widehat{X}$ *at* $P$ by

$$R_I(P, \widehat{X}) := \mathbb{E}_{X_{1:n} \sim P} \left[ \ell(P, P_{\widehat{X}(X_{1:n}, Z)|X_{1:n}}) \right].$$

We can then study the *minimax risk of sampling*, $M_I(\mathcal{F}_G, \ell, n) := \inf_{\widehat{X}} \sup_{P \in \mathcal{F}_G} R_I(P, \widehat{X})$. A few remarks about $M_I(\mathcal{F}, \ell, n)$: First, we implicitly assumed $\ell(P, P_{\widehat{X}(X_{1:n}, Z)|X_{1:n}})$ is well-defined, which is not obvious unless $P_{\widehat{X}(X_{1:n}, Z)} \in \mathcal{F}_G$. We discuss this assumption further below. Second, since the risk $R_I(P, \widehat{X})$ depends on the unknown true distribution $P$, we cannot calculate it in practice. Third, for the same reason (because $R_P(P, \widehat{X})$ depends directly on $P$ rather than particular data $X_{1:n}$), it detect lack-of-diversity issues such as mode collapse. As we discuss in the Appendix, these latter two points are distinctions from the recent work of [5] on generalization in GANs.

## 8.2 Comparison of Explicit and Implicit Generative Models

Algorithmically, sampling is a very distinct problem from density estimation; for example, many computationally efficient Monte Carlo samplers rely on the fact that a function *proportional* to the density of interest can be computed much more quickly than the exact (normalized) density function [11]. In this section, we show that, given unlimited computational resources, the problems of density estimation and sampling are equivalent in a minimax statistical sense. Since exactly minimax estimators ($\operatorname{argmin}_{\widehat{P}} \sup_{P \in \mathcal{F}_G} R_D(P, \widehat{P})$) often need not exist, the following weaker notion is useful for stating our results:

**Definition 8** (Nearly Minimax Sequence). A sequence $\{\widehat{P}_k\}_{k \in \mathbb{N}}$ of density estimators (resp., $\{\widehat{X}_k\}_{k \in \mathbb{N}}$ of generators) is called *nearly minimax over* $\mathcal{F}_G$ if $\lim_{k \to \infty} \sup_{P \in \mathcal{F}_G} R_{P,D}(\widehat{P}_k) = M_D(\mathcal{F}_G, \ell, n)$ (resp., $\lim_{k \to \infty} \sup_{P \in \mathcal{F}_G} R_{P,I}(\widehat{X}_k) = M_I(\mathcal{F}_G, \ell, n)$).

The following theorem identifies sufficient conditions under which, in the statistical minimax framework described above, density estimation is no harder than sampling. The idea behind the proof is as follows: If we have a good sampler $\widehat{X}$ (i.e., with $R_I(\widehat{X})$ small), then we can draw $m$ 'fake' samples from $\widehat{X}$. We can use these 'fake' samples to construct a density estimate $\widehat{P}$ of the implicit distribution of $\widehat{X}$ such that, under the technical assumptions below, $R_D(\widehat{P}) - R_I(\widehat{X}) \to 0$ as $m \to \infty$.

**Theorem 9** (Conditions under which Density Estimation is Statistically no harder than Sampling). *Let $\mathcal{F}_G$ be a family of probability distributions on a sample space $\mathcal{X}$. Suppose*

*(A1) $\ell : \mathcal{P} \times \mathcal{P} \to [0, \infty]$ is non-negative, and there exists $C_\triangle > 0$ such that, for all $P_1, P_2, P_3 \in \mathcal{F}_G$, $\ell(P_1, P_3) \leq C_\triangle \left( \ell(P_1, P_2) + \ell(P_2, P_3) \right)$.*

*(A2) $M_D(\mathcal{F}_G, \ell, m) \to 0$ as $m \to \infty$.*

*(A3) For all $m \in \mathbb{N}$, we can draw $m$ IID samples $Z_1, ..., Z_m \overset{IID}{\sim} Q_Z$ of the latent variable $Z$.*

*(A4) there exists a nearly minimax sequence of samplers $\widehat{X}_k : \mathcal{X}^n \times \mathcal{Z} \to \mathcal{X}$ such that, for each $k \in \mathbb{N}$, almost surely over $X_{1:n}$, $P_{\widehat{X}_k(X_{1:n},Z)|X_{1:n}} \in \mathcal{F}_G$.*

*Then, $M_D(\mathcal{F}_G, \ell, n) \leq C_\triangle M_I(\mathcal{F}_G, \ell, n)$.*

Assumption (A1) is a generalization of the triangle inequality (and reduces to the triangle inequality when $C_\triangle = 1$). This weaker assumption applies, for example, when $\ell$ is the Jensen-Shannon divergence (with $C_\triangle = 2$) used in the original GAN formulation of [17], even though this does not satisfy the triangle inequality [15]). Assumption (A2) is equivalent to the existence of a uniformly $\ell$-risk-consistent estimator over $\mathcal{F}_G$, a standard property of most distribution classes $\mathcal{F}_G$ over which density estimation is studied (e.g., our Theorem 1). Assumption (A3) is a natural design criterion of implicit generative models; usually, $Q_Z$ is a simple parametric distribution such as a standard normal.

Finally, Assumption (A4) is the most mysterious, because, currently, little is known about the minimax theory of samplers when $\mathcal{F}_G$ is a large space. On one hand, since $M_I(\mathcal{F}_G, \ell, n)$ is an infimum over $\widehat{X}$, Theorem 9 continues to hold if we restrict the class of samplers (e.g., to those satisfying Assumption (A4) or those we can compute). On the other hand, even without restricting $\widehat{X}$, this assumption may not be too restrictive, because nearly minimax samplers are necessarily close to $P \in \mathcal{F}_G$. For example, if $\mathcal{F}_G$ contains only smooth distributions but $\widehat{X}$ is the trivial empirical sampler described above, then $\ell(P, P_{\widehat{X}})$ should be large and $\widehat{X}$ is unlikely to be minimax optimal.

Finally, in practice, we often do not know estimators that are nearly minimax for finite samples, but may have estimators that are rate-optimal (e.g., as given by Theorem 1), i.e., that satisfy

$$C := \limsup_{n \to \infty} \frac{\sup_{P \in \mathcal{F}_G} R_I(P, \widehat{X})}{M_I(\mathcal{F}_G, \ell, n)} < \infty.$$

Under this weaker assumption, it is straightforward to modify our proof to conclude that

$$\limsup_{n \to \infty} \frac{M_D(\mathcal{F}_G, \ell, n)}{M_I(\mathcal{F}_G, \ell, n)} \leq C_\triangle C.$$

The converse result ($M_D(\mathcal{F}_G, \ell, n) \geq M_I(\mathcal{F}_G, \ell, n)$) is simple to prove in many cases, and is related to the well-studied problem of Monte Carlo sampling [38]; we discuss this briefly in the Appendix.

# 9  Conclusions

Given the recent popularity of implicit generative models in many applications, it is important to theoretically understand why these models appear to outperform classical methods for similar problems. This paper provided new minimax bounds for density estimation under adversarial losses, both with and without adaptivity to smoothness, and gave several applications, including both traditional statistical settings and perfectly optimized GANs. We also gave simple conditions under which minimax bounds for density estimation imply bounds for the problem of implicit generative modeling, suggesting that sampling is typically not *statistically* easier than density estimation. Thus, for example, the strong curse of dimensionality that is known to afflict to nonparametric density estimation Wasserman [49] should also limit the performance of implicit generative models such as GANs. The Appendix describes several specific avenues for further investigation, including whether the curse of dimensionality can be avoided when data lie on a low-dimensional manifold. zzz

**Acknowledgments**

This work was partly supported by NSF grant IIS1563887, the Darpa D3M program, AFRL FA8750-17-2-0212, and the NSF Graduate Research Fellowship DGE-1252522.

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
