[Supplementary Material]

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

# 10 Further Related Work

As noted in the main paper, our problem setting is quite general, and thus overlaps with several previous settings that have been studied. First, we note the analysis of [35], which also studied convergence of distribution estimation under adversarial losses. Considering a somewhat broader class of non-metric losses (including, e.g., Jensen-Shannon divergence), which they call *adversarial divergences*, [35] provided consistency results (in distribution) for a number of GAN formulations, assuming convergence of the min-max GAN optimization problem to a generator-optimal equilibrium. However, they did not study rates of convergence.

Our results can also be viewed as a refinement of several results from empirical process and learning theory, especially the wealth of literature on the case where $\mathcal{F}_D$ is a Glivenko-Cantelli (GC, a.k.a., Vapnik-Chervonenkis (VC)) class [44]. Corollary 2 can be interpreted as showing that spaces $\mathcal{F}_D$ that are sufficiently small in terms of orthonormal basis expansions are $n^{-1/2}$-uniformly GC/VC classes [2, 62]. In particular, this gives a simple functional-analytic proof of this property for the general case when $\mathcal{F}_D$ is a ball in a translation-invariant RKHS. On the other hand, some related results, cast in terms of fat-shattering dimensions [37, 18], appear to lead to slower rates for RKHSs.

Glivenko-Cantelli classes are defined without regards to the class $\mathcal{F}_G$ of possible distributions. However, the more interesting consequences of our results are for the case that $\mathcal{F}_G$ is restricted, as in Theorem 1. In Example 4 this allowed us to characterize the interaction between smoothness constraints on the discriminator class $\mathcal{F}_D$ and the generator class $\mathcal{F}_G$, showing in particular, that, when $\mathcal{F}_D$ is large, restricting $\mathcal{F}_G$ improves convergence rates. Aside for the results of [33] and many results for the specific case $\mathcal{F}_D = \mathcal{L}_\lambda^2$, we do not know of any results that show this.

Several prior works have studied the closely related problem of estimating certain adversarial metrics, including $\mathcal{L}^2$ distance [29], MMD [23], Sobolev distances [51], and others [56]. In some cases, these metrics can themselves be estimated far more efficiently than the underlying distribution under that loss, and these estimators have various applications including two-sample/homogeneity and independence testing [3, 23, 46], and distributional [58], transfer [16], and transductive [45] learning.

There has also been some work studying the min-max optimization problem in terms of which GANs are typically cast [41, 34]. However, in this work, as in [35, 33], we implicitly assume the optimization procedure has converged to a generator-optimal equilibrium. Another work that studies adversarial losses is [10], which focuses on a comparison of Wasserstein distance and MMD in the context of implicit generative modeling.

## 10.1 Other statistical analyses of GANs

Our results are closely related to some previous work studying the *generalization error* of GANs under MMD [18] or Jensen-Shannon divergence, Wasserstein, or other adversarial losses [7].

Assume, for simplicity, that $\ell$ satisfies a weak triangle inequality (Assumption (A1) above), and let $P$ denote the true distribution from which the data are drawn IID. Then, we can bound the true loss $\ell(P, \widehat{P})$ of an estimator $\widehat{P}$ in terms of the approximation error $\ell(P, P_*)$ (corresponding to bias) and generalization error $\ell(P_*, \widehat{P})$ (i.e., corresponding to variance):

$$\ell(P, \widehat{P}) \leq C_\triangle \left( \ell(P, P_*) + \ell(P_*, \widehat{P}) \right),$$

where $P_* := \mathrm{argmin}_{Q \in \widehat{\mathcal{F}}} \ell(P, Q)$ denotes the optimal approximation of $P$ in some restricted class $\widehat{\mathcal{F}} \subseteq \mathcal{F}_G$ of estimators in which $\widehat{P}$ lies.

Bounding the approximation error $\ell(P, P_*)$ typically requires restricting the space $\mathcal{F}_G$ in which $P$ lies. Theorem 1 of [18] and Theorem 3.1 of [7] focus on bounding the generalization error $\ell(P_*, \widehat{P})$, and thus avoid making such assumptions on $P$. However, our Theorem 1 shows that, when $\mathcal{F}_D$ is sufficiently small (e.g., an RKHS, as in [18]), $\ell = d_{\mathcal{F}_D}$ is so weak that $\ell(P, P_*)$ can be bounded even when $\mathcal{F}_G$ includes *all* probability measures. In particular, while [18] gave only high-probability bounds of order $n^{-1/2}$ on the *generalization error* $\ell(P_*, \widehat{P})$ in terms of the fat-shattering dimension of the RKHS, we show that, for any RKHS with a translation-invariant kernel, the *total* risk $\mathbb{E}[\ell(P, \widehat{P})]$ can be bounded at the parametric rate of $n^{-1/2}$.

[7] also showed that, if $\widehat{\mathcal{F}}$ is too large (specifically, if $\widehat{\mathcal{F}}$ contains the empirical distribution), then the generalization error $\ell(P_*, \widehat{P})$ (or, specifically, an empirical estimate thereof) need not vanish as the sample size increases, or, in the case of Wasserstein distance, if the dimension $d$ grows faster than logarithmically with the sample size $n$. Our Theorem 1 showed that, if $\widehat{F}$ contains only (e.g., orthogonal series) estimates of a fixed smoothness (e.g., orthogonal series estimates with a fixed $\zeta$), then the generalization error decays at the rate $\asymp \zeta^{d/2} n^{-1/2}$ (the first term on the right-hand side of 4), so that $d \in o(\log n)$ is still necessary[1]. Our minimax lower bound 3 suggests that, without making significantly stronger assumptions, we cannot hope to avoid this curse of dimensionality, at least without sacrificing approximation error (bias).

## 11    Proof of Upper Bound

In this section, we prove our main upper bound, Theorem 1. We begin with a simple lemma showing that, under mild assumptions, we can write an adversarial loss in terms of an $\mathcal{L}_\lambda^2$ basis expansion.

**Lemma 10** (Basis Expansion of Adversarial Loss). *Consider a class $\mathcal{F}_D$ of discriminator functions, two probability distributions $P$ and $Q$, and an orthonormal basis $\{\phi_z\}_{z \in \mathcal{Z}}$ of $\mathcal{L}_\lambda^2(\mathcal{X})$. Moreover, suppose that either of the following conditions holds:*

1. *$P, Q \ll \lambda$ have densities $p, q \in \mathcal{L}_\lambda^2$.*

2. *For every $f \in \mathcal{F}_D$, the expansion of $f$ in the basis $\mathcal{B}$ converges uniformly (over $\mathcal{X}$) to $f$. That is,*

$$\lim_{Z \uparrow \mathcal{Z}} \sup_{x \in \mathcal{X}} \left| f(x) - \sum_{z \in Z} \widetilde{f}_z(x) \phi_z(x) \right| \to 0.$$

*Then, we can expand the adversarial loss $d_{\mathcal{F}_D}$ over $\mathcal{P}$ as*

$$d_{\mathcal{F}_D}(P, Q) = \sup_{f \in \mathcal{F}_D} \sum_{z \in \mathcal{Z}} \widetilde{f}_z \left( \widetilde{P}_z - \widetilde{Q}_z \right).$$

Condition 1 above is quite straightforward, and would be taken for granted in most classical non-parametric analysis. When $\mathcal{B}$ is the Fourier basis, the assumption that $p, q \in \mathcal{L}_\mu^r$ for $r = 2$ can be weakened to any $r > 1$ using Hölder's inequality together with the facts that $f \in \mathcal{L}^{r'}$ and that Fourier series converge in $\mathcal{L}^{r'}$ (where $r' = \frac{r}{r-1}$ denote the Hölder conjugate of $r$).

Since we are also interested in probability distributions that lack density functions, we provide the fairly mild Condition 2 as an alternative. As an example of this condition in the Fourier case, suppose $\mathcal{F}_D$ is uniformly equi-continuous, say, with modulus of continuity $\omega : [0, \infty) \to [0, \infty)$ satisfying $\omega(\epsilon) \in o\left(\frac{1}{\log 1/\epsilon}\right)$. Then, there exists a constant $C > 0$ such that

$$\sup_{x \in \mathcal{X}} \left| f(x) - \sum_{|z| \leq \zeta} \widetilde{f}_z \phi_z(x) \right| \leq K(\log \zeta) \omega \left( \frac{2\pi}{\zeta} \right). \tag{9}$$

As a concrete example of this, it suffices if every $f$ is $\alpha_f$-Hölder continuous for some $\alpha_f > 0$. Finally, we note that, if $P$ and $Q$ are allowed to be arbitrary, then the above uniform convergence assumption is essentially also necessary.

*Proof.* First note that it suffices to show that, for all $f \in \mathcal{F}_D$,

$$\mathbb{E}_{X \sim P}[f(X)] - \mathbb{E}_{X \sim Q}[f(X)] = \sum_{z \in \mathcal{Z}} \widetilde{f}_z \left( \widetilde{P}_z - \widetilde{Q}_z \right).$$

We show this separately for the two sets of assumptions considered:

1. **Case 1: $P, Q$ have a densities $p, q \in \mathcal{L}_\mu^2$.** Then $\widetilde{P}_z = \langle p, \phi_z \rangle_{\mathcal{L}^2}$, and so, by the Plancherel Theorem, since $f \in \mathcal{L}_\mu^\infty(\mathcal{X}) \subseteq \mathcal{L}_\mu^2(\mathcal{X})$,

$$\mathbb{E}_{X \sim P}[f(X)] = \int_\mathcal{X} fp \, d\mu = \langle f, p \rangle_{\mathcal{L}_\mu^2} = \sum_{z \in \mathcal{Z}} \widetilde{f}_z \widetilde{P}_z < \infty.$$

Similarly, $\mathbb{E}_{X \sim Q}[f(X)] = \sum_{z \in \mathcal{Z}} \widetilde{f}_z \widetilde{Q}_z < \infty$. Since these quantities are finite, we can split the sum of differences

$$\sum_{z \in \mathcal{Z}} \widetilde{f}_z \left( \widetilde{P}_z - \widetilde{Q}_z \right) = \sum_{z \in \mathcal{Z}} \widetilde{f}_z \widetilde{P}_z - \sum_{z \in \mathcal{Z}} \widetilde{f}_z \widetilde{Q}_z = \mathbb{E}_{X \sim P}[f(X)] - \mathbb{E}_{X \sim Q}[f(X)].$$

2. **Case 2: For every $f \in \mathcal{F}_D$, the basis expansion of $f$ in $\mathcal{B}$ converges uniformly (over $\mathcal{X}$) to $f$.** Then,

$$\left| \mathbb{E}_{X \sim P}[f(X)] - \mathbb{E}_{X \sim Q}[f(X)] - \sum_{|z| \leq \zeta} \widetilde{f}_z \left( \widetilde{P}_z - \widetilde{Q}_z \right) \right|$$

$$= \left| \int_\mathcal{X} f(x) \, dP - \int_\mathcal{X} f(x) \, dQ - \sum_{|z| \leq \zeta} \widetilde{f}_z \left( \int_\mathcal{X} \phi_z(x) \, dP - \int_\mathcal{X} \phi_z(x) \, dQ \right) \right|$$

$$= \left| \int_\mathcal{X} f(x) \, dP - \int_\mathcal{X} f(x) \, dQ - \int_\mathcal{X} \sum_{|z| \leq \zeta} \widetilde{f}_z \phi_z(x) \, dP - \int_\mathcal{X} \sum_{|z| \leq \zeta} \widetilde{f}_z \phi_z(x) \, dQ \right|$$

$$= \left| \int_\mathcal{X} f(x) - \sum_{|z| \leq \zeta} \widetilde{f}_z \phi_z(x) \, dP + \int_\mathcal{X} f(x) - \sum_{|z| \leq \zeta} \widetilde{f}_z \phi_z(x) \, dQ \right|$$

$$\leq \int_\mathcal{X} \left| f(x) - \sum_{|z| \leq \zeta} \widetilde{f}_z \phi_z(x) \right| \, dP + \int_\mathcal{X} \left| f(x) - \sum_{|z| \leq \zeta} \widetilde{f}_z \phi_z(x) \right| \, dQ$$

$$\leq 2 \sup_{x \in \mathcal{X}} \left| f(x) - \sum_{|z| \leq \zeta} \widetilde{f}_z \phi_z(x) \right| \to 0 \quad \text{as } \zeta \to \infty.$$

$\square$

*Theorem* 1. Suppose that $\mu(\mathcal{X}) < \infty$ and there exist constants $L_D, L_G > 0$, real-valued nets $\{a_z\}_{z \in \mathcal{Z}}, \{b_z\}_{z \in \mathcal{Z}}$ such that $\mathcal{F}_D = \mathcal{H}_{p,a}(\mathcal{X}, L_D)$ and $\mathcal{F}_G = \mathcal{H}_{q,b}(\mathcal{X}, L_G)$, where $p, q \geq 1$. Let $p' = \frac{p}{p-1}$ denote the Hölder conjugate of $p$. Then, for any $P \in \mathcal{F}_G$,

$$\mathbb{E}_{X_{1:n}} \left[ d_{\mathcal{F}_D} \left( P, \widehat{P} \right) \right] \leq L_D \frac{c_{p'}}{\sqrt{n}} \left\| \left\{ \frac{\|\phi_z\|_{\mathcal{L}_P^\infty}}{a_z} \right\}_{z \in Z} \right\|_{p'} + L_D L_G \left\| \left\{ \frac{1}{a_z b_z} \right\}_{z \in \mathcal{Z} \setminus Z} \right\|_{1/(1-1/p-1/q)}.$$

*Proof.* By Lemma 10,

$$\mathbb{E}_{X_{1:n}} \left[ d_{\mathcal{F}_D} \left( P, \widehat{P} \right) \right] = \mathbb{E}_{X_{1:n}} \left[ \sup_{f \in \mathcal{F}_D} \sum_{z \in \mathcal{Z}} |\widetilde{f}_z \left( \widetilde{P}_z - \widehat{P}_z \right)| \right]$$

$$= \mathbb{E}_{X_{1:n}} \left[ \sup_{f \in \mathcal{F}_D} \sum_{z \in Z} |\widetilde{f}_z \left( \widetilde{P}_z - \widehat{P}_z \right)| + \sum_{z \in \mathcal{Z} \setminus Z} |\widetilde{f}_z \left( \widetilde{P}_z - \widehat{P}_z \right)| \right]$$

$$= \mathbb{E}_{X_{1:n}} \left[ \sup_{f \in \mathcal{F}_D} \sum_{z \in Z} |\widetilde{f}_z \left( \widetilde{P}_z - \widehat{P}_z \right)| + \sum_{z \in \mathcal{Z} \setminus Z} |\widetilde{f}_z \widetilde{P}_z| \right]$$

$$\leq \mathbb{E}_{X_{1:n}} \left[ \sup_{f \in \mathcal{F}_D} \sum_{z \in Z} |\widetilde{f}_z \left( \widetilde{P}_z - \widehat{P}_z \right)| \right] + \sup_{f \in \mathcal{F}_D} \sum_{z \in \mathcal{Z} \setminus Z} |\widetilde{f}_z \widetilde{P}_z|.$$

Note that we have decomposed the risk into two terms, the first comprising estimation error (variance) and the second comprising approximation error (bias). Indeed, in the case that $\mathcal{F}_D = \mathcal{L}^2(\mathcal{X})$, the above becomes precisely the usual bias-variance decomposition of mean squared error.

To bound the first term, applying the Holder's inequality, the fact that $f \in \mathcal{F}_D$, and Jensen's inequality (in that order), we have

$$
\begin{aligned}
\mathop{\mathbb{E}}_{X_{1:n}} \left[ \sup_{f \in \mathcal{F}_D} \sum_{z \in Z} |\tilde{f}_z \left( \tilde{P}_z - \hat{P}_z \right)| \right] &= \mathop{\mathbb{E}}_{X_{1:n}} \left[ \sup_{f \in \mathcal{F}_D} \sum_{z \in Z} a_z |\tilde{f}_z| \frac{|\tilde{P}_z - \hat{P}_z|}{a_z} \right] \\
&\leq \mathop{\mathbb{E}}_{X_{1:n}} \left[ \sup_{f \in \mathcal{F}_D} \left( \sum_{z \in Z} a_z^p |\tilde{f}_z|^p \right)^{\frac{1}{p}} \left( \sum_{z \in Z} \left( \frac{|\tilde{P}_z - \hat{P}_z|}{a_z} \right)^{p'} \right)^{\frac{1}{p'}} \right] \\
&\leq L_D \mathop{\mathbb{E}}_{X_{1:n}} \left[ \left( \sum_{z \in Z} \left( \frac{|\tilde{P}_z - \hat{P}_z|}{a_z} \right)^{p'} \right)^{\frac{1}{p'}} \right] \\
&\leq L_D \left( \sum_{z \in Z} \frac{\mathbb{E}_{X_{1:n}} \left[ \left| \tilde{P}_z - \hat{P}_z \right|^{p'} \right]}{a_z^{p'}} \right)^{\frac{1}{p'}} \leq \frac{L_D}{\sqrt{n}} \left( \sum_{z \in Z} \frac{\|\phi_z\|_{\mathcal{L}_P^\infty}^{p'}}{a_z^{p'}} \right)^{\frac{1}{p'}},
\end{aligned}
$$

where $p' = \frac{p}{p-1}$ is the Hölder conjugate of $p$. In the last inequality we have used Rosenthal's inequality i.e.,

$$
\mathop{\mathbb{E}}_{X_{1:n}} \left[ \left| \tilde{P}_z - \hat{P}_z \right|^{p'} \right] \leq c_{p'} \frac{\|\phi_z\|_{\mathcal{L}_P^\infty}^{p'}}{n^{p'/2}}.
$$

For the second term, by Holder's inequality,

$$
\begin{aligned}
\sup_{f \in \mathcal{F}_D} \sum_{z \in \mathcal{Z} \backslash Z} |\tilde{f}_z \tilde{P}_z| &\leq \sup_{f \in \mathcal{F}_D} \left( \sum_{z \in \mathcal{Z} \backslash Z} \left( a_z |\tilde{f}_z| \right)^p \right)^{1/p} \left( \sum_{z \in \mathcal{Z} \backslash Z} \left( \frac{|\tilde{P}_z|}{a_z} \right)^{p'} \right)^{1/p'} \\
&\leq L_D \left\| \left\{ \frac{b_z \tilde{P}_z}{b_z a_z} \right\}_{z \in \mathcal{Z} \backslash Z} \right\|_{p'} \\
&\leq L_D \left\| \{b_z \tilde{P}_z\}_{z \in \mathcal{Z} \backslash Z} \right\|_q \left\| \left\{ \frac{1}{b_z a_z} \right\}_{z \in \mathcal{Z} \backslash Z} \right\|_{\frac{p'q}{q-p'}} \quad \text{by Holder} \\
&= L_D L_G \left\| \left\{ \frac{1}{a_z b_z} \right\}_{z \in \mathcal{Z} \backslash Z} \right\|_{\frac{1}{1-(1/p+1/q)}}
\end{aligned}
$$

$\square$

## 12  Proof of Lower Bound

*Theorem* 3 (Minimax Lower Bound). Let $\lambda(\mathcal{X}) = 1$, and let $p_0$ denote the uniform density (with respect to Lebesgue measure) on $\mathcal{X}$. Suppose $\{p_0\} \cup \{\phi_z\}_{z \in \mathcal{Z}}$ is an orthonormal basis in $\mathcal{L}_\lambda^2$, suppose $\{a_z\}_{z \in \mathcal{Z}}$ and $\{b_z\}_{z \in \mathcal{Z}}$ are two real-valued nets, and let $L_D, L_G \geq 0$. For any $Z \subseteq \mathcal{Z}$, define

$$
A_Z := |Z|^{1/p} \sup_{z \in Z} a_z \quad \text{and} \quad B_Z := |Z|^{1/q} \sup_{z \in Z} b_z.
$$

Then, for $\mathcal{H}_D = \mathcal{H}_{p,a}(L_D)$ and $\mathcal{H}_G := \mathcal{H}_{b,q}(L_G)$, for any $Z \subseteq \mathcal{Z}$ satisfying

$$
B_Z \geq 16 L_G \sqrt{\frac{n}{\log 2}} \tag{10}
$$

and

$$2\frac{L_G}{B_Z}\sum_{z\in Z}\|\phi_z\|_{\mathcal{L}_\mu^\infty}\leq 1, \tag{11}$$

we have

$$M(\mathcal{H}_D,\mathcal{H}_G)\geq\frac{L_G L_D|Z|}{64A_Z B_Z}=\frac{L_G L_D|Z|^{1-1/p-1/q}}{64\left(\sup_{z\in Z}a_z\right)\left(\sup_{z\in Z}b_z\right)}.$$

*Proof.* We will follow a standard procedure for proving minimax lower bounds based on the Varshamov-Gilbert bound and Fano's lemma (as outlined, e.g., Chapter 2 of Tsybakov [61]). The proof is quite similar to a standard proof for the case of $\mathcal{L}_\lambda^2$-loss, based on constructing a finite "worst-case" subset $\Omega_G\subseteq\mathcal{F}_G$ of densities over which estimation is difficult. The main difference is that we also construct a similar finite "worst-case" subset $\Omega_D\subseteq\mathcal{F}_D$ of the discriminator class $\mathcal{F}_D$, which we use to lower bound $d_{\mathcal{F}_D}\geq d_{\Omega_D}$ over $\Omega_G$. Specifically, we will use the following result:

**Lemma 11** (Simplified Form of Theorem 2.5 of Tsybakov [61]). *Fix a family $\mathcal{P}$ of distributions over a sample space $\mathcal{X}$ and fix a pseudo-metric $\rho:\mathcal{P}\times\mathcal{P}\to[0,\infty]$ over $\mathcal{P}$. Suppose there exists a set $T\subseteq\mathcal{P}$ such that*

$$s:=\inf_{p,p'\in T}\rho(p,p')>0\quad and\quad\sup_{p\in T}D_{KL}(p,p_0)\leq\frac{\log|T|}{16},$$

*where $D_{KL}:\mathcal{P}\times\mathcal{P}\to[0,\infty]$ denotes Kullback-Leibler divergence. Then,*

$$\inf_{\widehat{p}}\sup_{p\in\mathcal{P}}\mathbb{E}\left[\rho(p,\widehat{p})\right]\geq\frac{s}{16},$$

*where the* inf *is taken over all estimators $\widehat{p}$ (i.e., (potentially randomized) functions of $\widehat{p}:\mathcal{X}\to\mathcal{P}$).*

Note that, compared to Theorem 2.5 of Tsybakov [61], we have loosened some of the constants in order to provide a simpler finite-sample statement.

Suppose $Z\subseteq\mathcal{Z}$ satisfies condition (10) and (11). For each $\tau\in\{-1,1\}^Z$ define

$$p_\tau:=p_0+c_G\sum_{z\in Z}\tau_z\phi_z,$$

where $c_G=\frac{L_G}{B_Z}$, and let $\Omega_G:=\left\{p_\tau:\tau\in\{-1,1\}^Z\right\}$.

Since each $\phi_z$ is orthogonal to $p_0$, each $p\in\Omega_G$ has unit mass $\int_{\mathcal{X}}p\,d\lambda=1$, and, by assumption (11),

$$\|p_\tau-p_0\|_{\mathcal{L}_\lambda^\infty}=\left\|\frac{L_G}{B_Z}\sum_{z\in Z}\tau_z\phi_z\right\|_{\mathcal{L}_\lambda^\infty}\leq\frac{L_G}{B_Z}\sum_{z\in Z}\|\phi_z\|_{\mathcal{L}_\lambda^\infty}\leq 0.5,$$

which implies that each $p\in\Omega_G$ is lower bounded on $\mathcal{X}$ by 0.5. Thus, each $p\in\Omega_G$ is a probability density. Note that, if we had worked with Gaussian sequences, as in Liang [33], we would not need to check this, and could hence omit assumption (11). Finally, by construction, for each $p\in\Omega_G$,

$$\|p\|_b^q=\sum_{z\in Z}b_z^q|p_z|^q=c^q\sum_{z\in Z}b_z^q\leq c^q|Z|\sup_{z\in Z}b_z^q=L_G^q$$

so that $\Omega_G\subseteq\mathcal{H}_{b,q}(L_G)$. Also, for $c_D:=\frac{L_D}{A_Z}$ and for each $\tau\in\{-1,1\}^Z$, let

$$f_\tau:=\frac{L_D}{A_Z}\sum_{z\in Z}\tau_z\phi_z,$$

and define $\Omega_D:=\left\{f_\tau:\tau\in\{-1,1\}^Z\right\}$. By construction, for each $f_\tau\in\Omega_D$,

$$\|f_\tau\|_a^p=\frac{L_D^p}{A_Z^p}\sum_{z\in Z}a_z^p\leq\frac{L_D^p}{A_Z^p}|Z|\sup_{z\in Z}a_z^p=L_D^p,$$

so that $\Omega_D \subseteq \mathcal{H}_{p,a}(L_D)$. Then, for any $\tau, \tau' \in \{-1, 1\}^Z$,

$$d_{\mathcal{F}_D}(p_\tau, p_{\tau'}) \geq d_{\Omega_D}(p_\tau, p_{\tau'}) = \sup_{\tau'' \in \{-1,1\}^Z} \sum_{z \in Z} f_{\tau'',z} c_G(\tau_z - \tau'_z) = 2c_G c_D \omega(\tau, \tau'),$$

where $\omega(\tau, \tau') := \sum_{z \in Z} 1_{\{\tau_z \neq \tau'_z\}}$ denotes the Hamming distance between $\tau$ and $\tau'$. By the Varshamov-Gilbert bound (Lemma 2.9 of Tsybakov [61]), we can select $T \subseteq \{-1, 1\}^Z$ such that $\log |T| \geq \frac{|Z| \log 2}{8}$ and, for each $\tau, \tau' \in T$,

$$\omega(\tau, \tau') \geq \frac{|Z|}{8}, \quad \text{so that} \quad d_{\mathcal{F}}(\theta_\tau, \theta_{\tau'}) \geq \frac{c_G c_D |Z|}{4}.$$

Moreover, for any $\tau \in \{-1, 1\}^Z$, using the facts that $-\log(1 + x) \leq x^2 - x$ for all $x \geq -0.5$ and that $\int_{\mathcal{X}} p_\tau \, dx = 1 = \int_{\mathcal{X}} p_0 \, dx$,

$$
\begin{aligned}
D_{KL}(p_\tau^n, p_0^n) &= n D_{KL}(p_\tau, p_0) \\
&= n \int_{\mathcal{X}} p_\tau(x) \log \frac{p_\tau(x)}{p_0(x)} \, dx \\
&= -n \int_{\mathcal{X}} p_\tau(x) \log \left( 1 + \frac{p_0(x) - p_\tau(x)}{p_\tau(x)} \right) dx \\
&\leq n \int_{\mathcal{X}} p_\tau(x) \left( \left( \frac{p_0(x) - p_\tau(x)}{p_\tau(x)} \right)^2 - \frac{p_0(x) - p_\tau(x)}{p_\tau(x)} \right) dx \\
&= n \int_{\mathcal{X}} \frac{(p_0(x) - p_\tau(x))^2}{p_\tau(x)} \, dx \\
&\leq 2n \int_{\mathcal{X}} (p_0(x) - p_\tau(x))^2 \, dx \\
&= 2n \|p_0 - p_\tau\|_{\mathcal{L}_\lambda^2}^2 = 2n \frac{L_G^2}{B_Z^2} |Z| \leq n \frac{L_G^2}{B_Z^2} \frac{16}{\log 2} \log |T| \leq \frac{\log |T|}{16},
\end{aligned}
$$

where the last two inequalities follow from the Varshamov-Gilbert bound and assumption (10), respectively. Combining the above results, Lemma 11 gives a minimax lower bound of

$$M(\mathcal{F}_D, \mathcal{F}_G) \geq \frac{c_G c_D |Z|}{64} = \frac{L_G L_D |Z|}{64 A_Z B_Z}.$$

$\square$

# 13  Proofs and Further Discussion of Applications in Section 6

*Example* 4 (Sobolev Spaces, Oracle and Adaptive estimators in Fourier basis). Suppose that, for some $s, t \geq 0$, $a_z = \left(1 + \|z\|_\infty^2\right)^{s/2}$ and $b_z = \left(1 + \|z\|_\infty^2\right)^{t/2}$. Then, one can check that, for $c = \frac{2^{d-2s} d}{d - 2s}$,

$$\sum_{z \in Z} a_z^{-2} \leq 1 + c\left(\zeta^{d-2s} - 1\right), \quad \sup_{z \in \mathcal{Z} \setminus Z} a_z^{-1} \leq \zeta^{-s}, \quad \text{and} \quad \sup_{z \in \mathcal{Z} \setminus Z} b_z^{-1} \leq \zeta^{-t},$$

so that Theorem 1 gives

$$\mathbb{E}_{X_{1:n}} \left[ d_{\mathcal{F}_D}\left(P, \widehat{P}\right) \right] \leq \frac{L_D}{\sqrt{n}} \left(1 + c\zeta^{d/2 - s}\right) + L_D L_G \zeta^{-(s+t)}. \tag{12}$$

Setting $\zeta = n^{\frac{1}{2t+d}}$ gives

$$\mathbb{E}_{X_{1:n}} \left[ d_{\mathcal{F}_D}\left(P, \widehat{P}\right) \right] \leq C n^{-\min\left\{ \frac{1}{2}, \frac{s+t}{2t+d} \right\}}, \quad \text{where} \quad C := L_D \left(2\sqrt{c} + L_G\right).$$

On the other hand, as long as $t > d/2$, setting

$$\zeta = \left( 256 L_G^2 \frac{n}{\log 2} \right)^{\frac{1}{2t+d}}$$

satisfies the conditions of Theorem 3, giving the minimax lower bound

$$M(\mathcal{W}^{s,2}, \mathcal{W}^{t,2}) \geq \frac{L_G L_D}{64\zeta^{s+t}} = c_1 n^{-\frac{s+t}{2t+d}} \quad \text{where} \quad c_1 = \frac{L_G L_D}{64} \left( \frac{\log 2}{256 L_G^2} \right)^{\frac{t+s}{2t+d}}.$$

Classical methods can also be used to show that, for all values of $s$ and $t$, $M(\mathcal{H}_{s,2}, \mathcal{H}_{t,2}) \geq c_2 n^{-1/2}$. Thus, we conclude, there exist constants $C, c > 0$ such that

$$cn^{-\min\left\{\frac{1}{2}, \frac{s+t}{2t+d}\right\}} \leq M\left(\mathcal{W}^{s,2}, \mathcal{W}^{t,2}\right) \leq Cn^{-\min\left\{\frac{1}{2}, \frac{s+t}{2t+d}\right\}}. \tag{13}$$

Combining the observation that the $s$-Hölder space $\mathcal{W}^{s,\infty} \subseteq \mathcal{W}^{s,2}$ with the lower bound in Theorem 3.1 of Liang [33], we have that (13) also holds when $\mathcal{H}_{s,2}$ is replaced with $\mathcal{W}^{s,\infty}$ (e.g., in the case of the Wasserstein metric $d_{\mathcal{W}^{1,\infty}}$), or indeed $\mathcal{W}^{s,q}$ for any $q \geq 2$.

*Corollary* 12 (Adaptive Upper Bound for Sobolev Spaces). *For any $t, \zeta \geq 0$ and $s \in (0, d/2)$,*

$$\sup_{P \in \mathcal{W}^{t,2}} \mathbb{E}_{X_{1:n} \overset{IID}{\sim} P} \left[ d_{\mathcal{W}^{s,2}}\left(P, \widehat{P}_{Z_\zeta}\right) \right] \leq C\zeta^{-s} \sup_{P \in \mathcal{W}^{t,2}} \mathbb{E}_{X_{1:n} \overset{IID}{\sim} P} \left[ d_{\mathcal{L}_\mu^2}\left(P, \widehat{P}_{Z_\zeta}\right) \right], \tag{14}$$

*where $C := \sqrt{2}\left(1 + \frac{2^{d-2s}d}{d-2s}\right)$ does not depend on $n$ or $\zeta$. Hence, if $\widehat{\zeta}(X_{1:n})$ is any adaptive scheme for choosing $\zeta$ (i.e., if computing $\widehat{\zeta}$ does not require knowledge of $t$), then $\widehat{P}_{\widehat{\zeta}}$ is adaptively minimax under the loss $d_{\mathcal{W}^{s,2}}$; that is, for all $t > 0$, there exists $C > 0$ such that*

$$\sup_{P \in \mathcal{W}^{t,2}} \mathbb{E}_{X_{1:n} \overset{IID}{\sim} P} \left[ d_{\mathcal{W}^{s,2}}\left(P, \widehat{P}_{Z_{\widehat{\zeta}}}\right) \right] \leq M\left(\mathcal{W}^{s,2}, \mathcal{W}^{t,2}\right).$$

*One common scheme for choosing $\widehat{\zeta}$ is to use a leave-one-out cross-validation scheme. Specifically, for*

$$\widehat{J}(\zeta) := \|\widehat{P}_\zeta\|_2^2 - \frac{2}{n} \sum_{i=1}^{n} \widehat{P}_{\zeta,-i}(X_i), \quad \text{where} \quad \widehat{P}_{\zeta,-i} := \sum_{z \in Z_\zeta} \left( \frac{1}{n-1} \sum_{j \in [n] \setminus \{i\}} \phi_z(X_j) \right) \phi_z$$

*is a computation of the estimate $\widehat{P}_\zeta$ omitting the $i^{th}$ sample $X_i$, one can show that $\mathbb{E}_{X_{1:n} \overset{IID}{\sim} P}\left[\widehat{J}(\zeta)\right] = \mathbb{E}_{X_{1:n} \overset{IID}{\sim} P}\left[d_{\mathcal{L}_\mu^2}^2\left(P, \widehat{P}_\zeta\right)\right] - \|P\|_{\mathcal{L}_\mu^2}^2$, so that, up to an additive constant independent of $\zeta$, $\widehat{J}(\zeta)$ is an unbiased estimate of the squared $\mathcal{L}_\mu^2$-risk using the parameter $\zeta$. Based on this, setting*

$$\widehat{\zeta} := \operatorname*{argmin}_{\zeta \in [0, n^{-1/d}]} J(\zeta),$$

*one can show that $\widehat{P}_{\widehat{\zeta}}$ is adaptively minimax over all Sobolev spaces $\mathcal{W}^{t,2}$ with $t > 0$; that is, for all $t > 0$,*

$$\sup_{P \in \mathcal{W}^{t,2}} \mathbb{E}_{X_{1:n} \overset{IID}{\sim} P} \left[ d_{\mathcal{L}_\mu^2}\left(P, \widehat{P}_{\widehat{\zeta}}\right) \right] \asymp M\left(\mathcal{L}_\mu^2, \mathcal{W}^{t,2}\right). \tag{15}$$

*This equivalence (14) implies that we can generalize the adaptive minimaxity bound (15) to*

$$\sup_{P \in \mathcal{W}^{t,2}} \mathbb{E}_{X_{1:n} \overset{IID}{\sim} P} \left[ d_{\mathcal{W}^{s,2}}\left(P, \widehat{P}_{\widehat{\zeta}}\right) \right] \asymp M\left(\mathcal{W}^{s,2}, \mathcal{W}^{t,2}\right). \tag{16}$$

*for all $s \in [0, d/2)$.*

*Proof.* A proof of the adaptive minimaxity of the cross-validation estimator in $d_{\mathcal{L}_\mu^2}$ can be found in Sections 7.2.1 and 7.5.1 of Massart [36]. Therefore, we prove only Inequality (14) here. To do this, we combine Theorem 1 with a lower bound on the worst-case performance of the orthogonal series estimator under $\mathcal{L}_\mu^2$ loss, which we establish by explicitly constructing a worst-case true distribution as follows.

Define $P_\zeta := 1 + L_G \zeta^{-t} \phi_\zeta$ (where $\phi_\zeta$ is any $\phi_z$ satisfying $\|z\|_\infty = \zeta$), one can easily check that $P_\zeta \in \mathcal{W}^{t,2}$, and that, for any $z$ with $\|z\| < \zeta$,

$$
\mathop{\mathbb{E}}_{X_{1:n} \overset{IID}{\sim} P_\zeta} \left[ \left( \widetilde{(P_\zeta)}_z - \widehat{P}_z \right)^2 \right] = \mathop{\mathbb{E}}_{X_{1:n} \overset{IID}{\sim} P_\zeta} \left[ \left( \frac{1}{n} \sum_{i=1}^n \phi_z(X_i) \right)^2 \right]
$$

$$
= \frac{1}{n} \mathop{\mathbb{E}}_{X \sim P_\zeta} \left[ \phi_z^2(X) \right]
$$

$$
= \frac{1}{n} \int_{\mathcal{X}} \phi_z^2(x) \left( 1 + L_G \zeta^{-t} \phi_\zeta(x) \right) \, dx
$$

$$
\geq \frac{1}{n} \int_{\mathcal{X}} \phi_z^2(x) \, dx = \frac{1}{n}
$$

(with equality if $\zeta \neq 2z$). Also, let

$$
f := \frac{L_D}{\sqrt{2}} \sum_{\|z\| < \zeta} \frac{\left( \widetilde{P_\zeta}_z - \widehat{P}_z \right)}{\sqrt{|Z_\zeta|}} \phi_z + \frac{L_D}{\sqrt{2}} \phi_\zeta
$$

so that

$$
\|f\|_2^2 = \frac{L_D^2}{2} \sum_{\|z\| < \zeta} \frac{\left( \widetilde{P_\zeta}_z - \widehat{P}_z \right)^2}{|Z_\zeta|} + \frac{L_D^2}{2} \leq \frac{L_D^2}{2} \sum_{\|z\| < \zeta} |Z_\zeta|^{-1} + \frac{L_D^2}{2} \leq L_D^2,
$$

and hence $f \in \mathcal{L}_\mu^2(1)$. Then,

$$
\mathop{\mathbb{E}}_{X_{1:n} \overset{IID}{\sim} P_\zeta} \left[ d_{\mathcal{L}_\mu^2} \left( P_\zeta, \widehat{P}_{Z_\zeta} \right) \right] \geq \mathop{\mathbb{E}}_{X_{1:n} \overset{IID}{\sim} P} \left[ \sum_{\|z\| < \zeta} \widetilde{f}_z \left( \widetilde{P_\zeta}_z - \widehat{P}_z \right)^2 + \widetilde{f}_\zeta \widetilde{P_\zeta}_z \right]
$$

$$
= \frac{L_D}{\sqrt{2|Z_\zeta|}} \sum_{\|z\| < \zeta} \mathop{\mathbb{E}}_{X_{1:n} \overset{IID}{\sim} P} \left[ \left( \widetilde{P_\zeta}_z - \widehat{P}_z \right)^2 \right] + \frac{L_D L_G}{\sqrt{2}} \zeta^{-t}
$$

$$
\geq \frac{L_D}{\sqrt{2|Z_\zeta|}} \sum_{\|z\| < \zeta} \frac{1}{\sqrt{n}} + \frac{L_D L_G}{\sqrt{2}} \zeta^{-t} = \frac{L_D}{\sqrt{2}} \left( \sqrt{\frac{\zeta^d}{n}} + L_G \zeta^{-t} \right)
$$

It follows that

$$
\sup_{P \in \mathcal{W}^{t,2}} \mathop{\mathbb{E}}_{X_{1:n} \overset{IID}{\sim} P} \left[ d_{\mathcal{L}_\mu^2} \left( P, \widehat{P}_{Z_\zeta} \right) \right] \geq \frac{L_D}{\sqrt{2}} \left( \sqrt{\frac{\zeta^d}{n}} + \zeta^{-t} \right).
$$

On the other hand, as we already saw, Theorem 1 gives

$$
\sup_{P \in \mathcal{W}^{t,2}} \mathop{\mathbb{E}}_{X_{1:n}} \left[ d_{\mathcal{W}^{s,2}} \left( P, \widehat{P} \right) \right] \leq \left( 1 + \frac{2^{d-2s} d}{d - 2s} \right) L_D \left( \sqrt{\frac{\zeta^d}{n}} + L_G \zeta^{-t} \right) \zeta^{-s}.
$$

Combining these two inequalities gives

$$
\sup_{P \in \mathcal{W}^{t,2}} \mathop{\mathbb{E}}_{X_{1:n}} \left[ d_{\mathcal{W}^{s,2}} \left( P, \widehat{P} \right) \right] \leq C \zeta^{-s} \sup_{P \in \mathcal{W}^{t,2}} \mathop{\mathbb{E}}_{X_{1:n} \overset{IID}{\sim} P} \left[ d_{\mathcal{L}_\mu^2} \left( P, \widehat{P}_{Z_\zeta} \right) \right].
$$

$\square$

## 13.1 Wavelet Basis

Our previous applications were given in terms of the Fourier basis. In this section, we demonstrate that our upper and lower bounds can give tight minimax results using other bases (in this case, the Haar wavelet basis).

Suppose that $\mathcal{X} = [0,1]^D$, and suppose that a function $f : \mathcal{X} \to \mathbb{R}$ has Haar wavelet basis coefficients $\widetilde{f}_{i,j}$, indexed by $z \in \mathcal{Z} := \{(i,j) \in \mathbb{N} \times \mathbb{N} : j \in [2^i]\}$, where $i \in \mathbb{N}$ is the order and $j \in [2^i]$ is the index within that order.

One can show (see, e.g., Donoho et al. [15]) that the Besov seminorm $\|\cdot\|_{\mathcal{B}_{p,q}^q}$ satisfies

$$\|f\|_{\mathcal{B}_{p,q}^r}^q = \sum_{i \in \mathbb{N}} 2^{iqs} \left( \sum_{j \in [2^i]} |\widetilde{f}_{i,j}|^p \right)^{q/p} = \sum_{i \in \mathbb{N}} 2^{iqs} \|\widetilde{f}_i\|_p^q,$$

where $s = r + \frac{1}{2} - \frac{1}{p}$. In particular, when $p = q = 2$, $s = r$, and one can show that $\mathcal{B}_{p,q}^r = \mathcal{W}_2^r$, and

$$\|f\|_{\mathcal{B}_{p,q}^r}^q = \sum_{(i,j) \in \mathcal{Z}} 2^{2is} |\widetilde{f}_{i,j}|^2,$$

For some $\zeta > 0$, we will choose the truncation set $Z$ to be of the form

$$Z = \{(i,j) \in \mathcal{Z} : i \leq \zeta\}.$$

Note that, for each $i \in \mathbb{N}$, since $\phi_{i,1}, ..., \phi_{i,2^i}$ have disjoint supports

$$\sup_{x \in \mathcal{X}} \sum_{j \in [2^i]} |\phi_{i,j}(x)| = \sup_{x \in \mathcal{X}} \sup_{j \in [2^i]} |\phi_{i,j}(x)| = 2^{i/2}.$$

Thus,

$$\sum_{j \in [2^i]} \|\phi_{i,j}\|_{\mathcal{L}_P^2}^2 = \sum_{j \in [2^i]} \int_{\mathcal{X}} \phi_{i,j}^2(x) dP \leq \int_{\mathcal{X}} \left( \sum_{j \in [2^i]} \phi_{i,j}(x) \right)^2 dP = 2^i.$$

**Example 13** (Sobolev Space, Wavelet Basis). Suppose that, for some $s, t \geq 0$, $a_{i,j} = 2^{is}$ and $b_{i,j} = 2^{it}$. Then, one can check that, for some $c > 0$

$$\sum_{z \in \mathcal{Z}} \frac{\|\phi_z\|_{\mathcal{L}_P^2}^2}{a_z^2} = \sum_{i \leq \zeta} \sum_{j \in [2^i]} \frac{\|\phi_{i,j}\|_{\mathcal{L}_P^2}^2}{2^{2is}} = \sum_{i \leq \zeta} \frac{2^i}{2^{2is}} = \frac{2^{(\zeta+1)(1-2s)} - 1}{2^{1-2s} - 1} \asymp 2^{\zeta(1-2s)}.$$

Also, $\sup_{z \in \mathcal{Z} \backslash Z} a_z^{-1} \leq 2^{-s\zeta}$ and $\sup_{z \in \mathcal{Z} \backslash Z} b_z^{-1} \leq 2^{-t\zeta}$. Thus, Theorem 1 gives

$$\mathbb{E}_{X_{1:n}} \left[ d_{\mathcal{F}_D} \left( P, \widehat{P} \right) \right] \lesssim L_D \left( \sqrt{\frac{c}{n}} 2^{(d/2-s)\zeta} + L_G 2^{-(s+t)\zeta} \right).$$

By letting $\zeta = \log_2 \xi$, we can easily see that this is identical, up to constants, to the bound for the Sobolev case. In contrast to Fourier basis, a larger variety of function spaces (such as inhomogeneous Besov spaces) can be expressed in terms of wavelet basis. The classical work of Donoho et al. [15] showed that, under $\mathcal{L}_\mu^p$ losses, linear estimators, such as that analyzed in our Theorem 1 are sub-optimal in these spaces, but that relatively simple thresholding estimators can recover the minimax rate. We leave it to future work to understand how this phenomenon extends to more general adversarial losses.

# 14 Proofs and Applications of Explicit & Implicit Generative Modeling Results (Section 8 of Main Paper)

Here, we prove Theorem 9 from the main text, provide some discussion of when the converse direction $M_I(\mathcal{P}, \ell, n) \leq M_D(\mathcal{P}, \ell, n)$ holds, and also provide some concrete applications.

## 14.1 Proofs of Theorem 9 and Converse

*Theorem* 9 (Conditions under which Density Estimation is Statistically no harder than Sampling). Let $\mathcal{F}_G$ be a family of probability distributions on a sample space $\mathcal{X}$. Assume the following:

**(A1)** $\ell : \mathcal{P} \times \mathcal{P} \to [0, \infty]$ is non-negative, and there exists $C_\triangle > 0$ such that, for all $P_1, P_2, P_3 \in \mathcal{F}_G$,

$$\ell(P_1, P_3) \leq C_\triangle \left( \ell(P_1, P_2) + \ell(P_2, P_3) \right).$$

**(A2)** $M_D(\mathcal{F}_G, \ell, m) \to 0$ as $m \to \infty$.

**(A3)** For all $m \in \mathbb{N}$, we can draw $m$ IID samples $Z_{1:m} = Z_1, ..., Z_m \overset{IID}{\sim} Q_Z$ of the latent variable $Z$.

**(A4)** there exists a nearly minimax sequence of samplers $\widehat{X}_k : \mathcal{X}^n \times \mathcal{Z} \to \mathcal{X}$ such that, for each $k \in \mathbb{N}$, almost surely over $X_{1:n}$, $P_{\widehat{X}_k(X_{1:n}, Z)|X_{1:n}} \in \mathcal{F}_G$.

Then, $M_D(\mathcal{F}_G, \ell, n) \leq C_\triangle M_I(\mathcal{F}_G, \ell, n)$.

*Proof.* The assumption (A2) implies that there exists a sequence $\{\widehat{P}_m\}_{m\in\mathbb{N}}$ of density estimators $\widehat{P}_m : \mathcal{X}^m \to \mathcal{P}$ that is uniformly consistent in $\ell$ over $\mathcal{P}$; that is,

$$\lim_{m\to\infty} \sup_{P\in\mathcal{P}} \mathbb{E}_{Y_{1:m} \overset{IID}{\sim} P} \left[ \ell \left( P, \widehat{P}_m(Y_{1:m}) \right) \right]. \tag{17}$$

For brevity, we use the abbreviation $P_{\widehat{X}_k} = P_{\widehat{X}_k(X_{1:n}, Z)|X_{1:n}}$ in the rest of this proof to denote the conditional distribution of the 'fake data' generated by $\widehat{X}_k$ given the true data. Recalling that the minimax risk is at most the risk of any particular sampler, we have

$$M_D(\mathcal{P}, \ell, n) := \inf_{\widehat{P}} \sup_{P\in\mathcal{P}} \mathbb{E}_{\substack{X_{1:n} \overset{IID}{\sim} P \\ Z_{1:m} \overset{IID}{\sim} Q_Z}} \left[ \ell \left( P, \widehat{P}(X_{1:n}) \right) \right]$$

$$\leq \sup_{P\in\mathcal{P}} \mathbb{E}_{\substack{X_{1:n} \overset{IID}{\sim} P \\ Z_{1:m} \overset{IID}{\sim} Q_Z}} \left[ \ell \left( P, \widehat{P}_m(X_{n+1:n+m}) \right) \right].$$

Taking $\lim_{m\to\infty}$ gives, by Tonelli's theorem and non-negativity of $\ell$,

$$M_D(\mathcal{P}, \ell, n)$$

$$\leq \lim_{m\to\infty} \sup_{P\in\mathcal{P}} \mathbb{E}_{\substack{X_{1:n} \overset{IID}{\sim} P \\ Z_{1:m} \overset{IID}{\sim} Q_Z}} \left[ \ell \left( P, \widehat{P}_m(X_{n+1:n+m}) \right) \right]$$

$$\leq C_\triangle \lim_{m\to\infty} \sup_{P\in\mathcal{P}} \mathbb{E}_{\substack{X_{1:n} \overset{IID}{\sim} P \\ Z_{1:m} \overset{IID}{\sim} Q_Z}} \left[ \ell \left( P, P_{\widehat{X}_k} \right) + \ell \left( P_{\widehat{X}_k}, \widehat{P}_m(X_{n+1:n+m}) \right) \right]$$

$$\leq C_\triangle \lim_{m\to\infty} \sup_{P\in\mathcal{P}} \mathbb{E}_{\substack{X_{1:n} \overset{IID}{\sim} P \\ Z_{1:m} \overset{IID}{\sim} Q_Z}} \left[ \ell \left( P, P_{\widehat{X}_k} \right) + \ell \left( P_{\widehat{X}_k}, \widehat{P}_m(X_{n+1:n+m}) \right) \right]$$

$$\leq C_\triangle \sup_{P\in\mathcal{P}} \mathbb{E}_{X_{1:n} \overset{IID}{\sim} P} \left[ \ell \left( P, P_{\widehat{X}_k} \right) \right] \tag{18}$$

$$+ C_\triangle \lim_{m\to\infty} \sup_{P\in\mathcal{P}} \mathbb{E}_{\substack{X_{1:n} \overset{IID}{\sim} P \\ Z_{1:m} \overset{IID}{\sim} Q_Z}} \left[ \ell \left( P_{\widehat{X}_k}, \widehat{P}_m(X_{n+1:n+m}) \right) \right]. \tag{19}$$

In the above, we upper bounded $M_D(\mathcal{P}, \ell, n)$ by the sum of two terms, (18) and (19). Since the sequence $\{\widehat{X}_k\}_{k\in\mathbb{N}}$ is nearly minimax, if we were to take an infimum over $k \in \mathbb{N}$ on both sides, the term (18) would become precisely $C_\triangle M_I(\mathcal{P}, \ell, n)$. Therefore, it suffices to observe that the second term (19) is 0. Indeed, by the assumption that $P_{\widehat{X}_k} \in \mathcal{P}$ for all $X_{1:n} \in \mathcal{X}$ and the uniform

consistency assumption (17),

$$
\lim_{m \to \infty} \sup_{\substack{P \in \mathcal{P} \\ X_{1:n} \overset{IID}{\sim} P \\ Z_{1:m} \overset{IID}{\sim} Q_Z}} \mathbb{E} \left[ \ell \left( P_{\widehat{X}_k}, \widehat{P}_m(X_{n+1:n+m}) \right) \right]
$$

$$
\leq \lim_{m \to \infty} \sup_{P \in \mathcal{P}, X_{1:n} \overset{IID}{\sim} P} \mathbb{E}_{Z_{1:m} \overset{IID}{\sim} Q_Z} \left[ \ell \left( P_{\widehat{X}_k}, \widehat{P}_m(X_{n+1:n+m}) \right) \right]
$$

$$
\leq \lim_{m \to \infty} \sup_{\substack{P' \in \mathcal{P} \\ X_{n+1:n+m} \overset{IID}{\sim} P'}} \mathbb{E} \left[ \ell \left( P, \widehat{P}_m(X_{n+1:n+m}) \right) \right] = 0.
$$

$\square$

For completeness, we provide a very simple result on the converse of Theorem 9:

**Theorem 14** (Conditions under which Sampling is Statistically no harder than Density Estimation).
*Suppose that, there exists as nearly minimax sequence $\{\widehat{P}_k\}_{k \in \mathbb{N}}$ such that, for any $k \in \mathbb{N}$, we can draw a random sample $\widehat{X}$ from $\widehat{P}_k(X_{1:n})$. Then,*

$$
M_D(\mathcal{F}_G, \ell, n) \geq M_I(\mathcal{F}_G, \ell, n).
$$

The assumption above that we can draw samples from a nearly minimax sequence of estimators if not particularly insightful, but techniques for drawing such samples have been widely studied in the vast literature of Monte Carlo sampling [49]. As an example, if $\widehat{P}$ is a kernel density estimator with kernel $K$, then, recalling that $K$ is itself a probability density, of which $\widehat{P}$ is a mixture, we can sample from $\widehat{P}$ simply by choosing a sample uniformly from $X_{1:n}$ and adding noise $\epsilon \sim K$. Alternatively, if $\widehat{P}$ is bounded and has bounded support, then one can perform rejection sampling.

*Proof.* Since, by definition of the implicit distribution of $\widehat{X}$,

$$
P_{\widehat{X}(X_{1:n}, Z)|X_{1:n}} = \widehat{P}(X_{1:n})
$$

is precisely the implicit distribution of $\widehat{X}$, we trivially have

$$
M_I(\mathcal{F}_G, \ell, n) \leq \sup_{P \in \mathcal{F}_G} \mathbb{E}_{X_{1:n} \overset{IID}{\sim} P} \left[ \ell \left( P, P_{\widehat{X}(X_{1:n}, Z)|X_{1:n}} \right) \right]
$$

$\square$

## 14.2 Applications

**Example 15** (Density Estimation and Sampling in Sobolev families under Dual-Sobolev Loss). There exist constants $C > c > 0$ such that, for all $n \in \mathbb{N}$,

$$
cn^{-\min\left\{ \frac{s+t}{2s+d}, \frac{1}{2} \right\}} \leq M_I \left( \mathcal{W}^{t,2}, d_{\mathcal{W}^{s,2}}, n \right) \leq Cn^{-\min\left\{ \frac{s+t}{2s+d}, \frac{1}{2} \right\}}.
$$

*Proof.* Since adversarial losses always satisfy the triangle inequality, the first inequality follows Theorems 9 and the discussion in Example 4. For the second inequality, since we have already established that the orthogonal series estimator $\widehat{P}_Z$ is nearly minimax, by Theorem 14 it suffices to give a scheme for sampling from the distribution $\widehat{P}_Z(X_{1:n})$. Since the sample space $\mathcal{X} = [0, 1]^d$ is bounded and the estimator $\widehat{P}_Z(X_{1:n})$ has a bounded density $p : \mathcal{X} \to [0, \infty)$, we can simply perform rejection sampling; that is, repeatedly sample $Z \times Y$ uniformly from $\mathcal{X} \times [0, \sup_{x \in \mathcal{X}} p(x)]$. Let $Z^*$ denote the first $Z$ sample satisfying $Y < p(Z)$. Then, we $Z^*$ will necessarily have the density $p$. $\square$

**Example 16** (Density Estimation and Sampling in Exponential Families under Jensen-Shannon, $\mathcal{L}^q$, Hellinger, and RKHS losses). Let $\mathcal{H}$ be an RKHS over a compact sample space $\mathcal{X} \subseteq \mathbb{R}^d$, and let

$$\mathcal{F}_G := \left\{ p_f : \mathcal{X} \to [0, \infty) \,\Big|\, p_f(x) = e^{f(x)-A(f)} \text{ for all } x \in \mathcal{X}, f \in \mathcal{H} \right\},$$

in which $A(f) := \log \int_{\mathcal{X}} e^{f(x)} \, d\mu$ denotes the log-partition function.

The Jensen-Shannon divergence $J : \mathcal{P} \times \mathcal{P} \to [0, \infty]$ is defined by

$$J(P,Q) := \frac{1}{2} \left( D_{KL}\left( P, \frac{P+Q}{2} \right) + D_{KL}\left( Q, \frac{P+Q}{2} \right) \right),$$

where $\frac{P+Q}{2}$ denotes the uniform mixture of $P$ and $Q$, and, noting that we always have $P \ll \frac{P+Q}{2}$ and $Q \ll \frac{P+Q}{2}$,

$$D_{KL}(P,Q) := \int_{\mathcal{X}} \log \left( \frac{dP}{dQ} \right) dP$$

denotes the Kullback-Leibler divergence. Although $J$ does not satisfy the triangle inequality, one can show that $\sqrt{J}$ is a metric on $\mathcal{P}$ [20], and hence, for all $P, Q \in \mathcal{P}$, by Cauchy-Schwarz,

$$J(P,Q) = \left( \sqrt{J(P,Q)} \right)^2 \leq \left( \sqrt{J(P,R)} + \sqrt{J(R,Q)} \right)^2 \leq 2J(P,R) + 2J(R,Q). \qquad (20)$$

Also, under mild regularity conditions on $\mathcal{H}$, Sriperumbudur et al. [53] (in their Theorem 7) provides uniform convergence guarantees for a particular density estimator over $\mathcal{P}$. Combining this the inequality (20), our Theorem 9 implies

$$M_D(\mathcal{P}, J, n) \leq 2M_I(\mathcal{P}, J, n).$$

For the same class $\mathcal{P}$, the convergence results of Sriperumbudur et al. [53] (their Theorems 6 and 7) also imply similar guarantees under several other losses, including the parameter estimation loss $\|f_P - f_{\widehat{P}}\|_{\mathcal{H}}$ in the RKHS metric, as well as the $\mathcal{L}^q_\mu$ and Hellinger metrics $H$ (on the density), so that we have $M_D(\mathcal{P}, \rho, n) \leq M_I(\mathcal{P}, \rho, n)$ when $\rho$ is any of these metrics.

Perhaps more interestingly, in the case of Jensen-Shannon divergence, under certain regularity conditions, we can altogether drop the assumption that $P_{\widehat{X}_k(X_{1:n},Z)|X_{1:n}} \in \mathcal{P}$ using uniform convergence bounds shown in Section 5 of Sriperumbudur et al. [53] for the mis-specified case; the density estimator described therein converges (uniformly over $P_*$) to the projection $P_*$ of $P_{\widehat{X}_k(X_{1:n},Z)|X_{1:n}}$ onto $\mathcal{P}$ even when samples are drawn from $P_{\widehat{X}_k(X_{1:n},Z)|X_{1:n}}$.

It is also worth pointing out that, when densities in $\mathcal{F}_G$ are additionally assumed to be lower bounded by a positive constant $\kappa > 0$ (i.e.,

$$\kappa := \inf_{p \in \mathcal{F}_G} \inf_{x \in \mathcal{X}} p(x) > 0,$$

then, by the inequality $-\log(1+x) \leq x^2 - x$ that holds for all $x \geq -0.5$, for all densities $p, q \in \mathcal{F}_G$,

$$\begin{aligned}
\int_{\mathcal{X}} p(x) \log \left( \frac{2p(x)}{p(x)+q(x)} \right) dx &= -\int_{\mathcal{X}} p(x) \log \left( 1 + \frac{q(x)-p(x)}{2p(x)} \right) dx \\
&\leq \int_{\mathcal{X}} p(x) \left( \left( \frac{q(x)-p(x)}{2p(x)} \right)^2 - \left( \frac{q(x)-p(x)}{2p(x)} \right) \right) dx \\
&= \int_{\mathcal{X}} \frac{(q(x)-p(x))^2}{2p(x)} \, dx \leq \frac{1}{2\kappa} \|P-Q\|^2_{\mathcal{L}^2_\mu},
\end{aligned}$$

and, therefore, $J(P,Q) \leq \frac{1}{2\kappa} \|P-Q\|_{\mathcal{L}^2_\mu}$. Thus, under this additional assumption of uniform lower-boundedness, standard results for density estimation under $\mathcal{L}^2_\mu$ apply [61].

## 15 Experimental Results

This section presents some empirical results supporting the theoretical bounds above. First, we consider an example with a finite basis, which should yield the parametric $n^{-1/2}$ rate. In particular, we construct the true distribution $P$ to consist of 6 randomly chosen basis functions in the Fourier basis. We employ the truncated series estimator $\widehat{P}$ of (3) in the same basis using different number of samples $n$ and compute the distance $d_{\mathcal{F}_D}\left(P, \widehat{P}\right)$. Under this setting, the maximization

(a) Parametric Regime      (b) Nonparametric Regime

Figure 1: Simple synthetic experiment to showcase the tightness of the bound.

problem of (1) needed to evaluate this distance can be solved in closed form. The risk empirically appears to closely follow our derived minimax rate of $n^{-1/2}$, as shown in Figure 1a. Next, we consider a non-parametric case, in which the number of active basis elements increases as function of $n$, weighted such that Inequality (6) predicts a rate of $n^{-1/3}$. As expected, the estimated risk, shown in Figure 1b, closely resembles the rate of $n^{-1/3}$.

## 16 Future Work

In this paper, we showed that minimax convergence rates for distribution estimation under certain adversarial losses can improve when the probability distributions are assumed to be smooth, using an orthogonal series estimator that smooths the observed empirical distribution. On the other hand, recent work has also shown that, at least under Wasserstein losses, minimax convergence rates improve when the distribution is assumed to have support of low intrinsic dimension, even within a high-dimensional ambient space [50]. In any case, further work is needed to understand whether minimax rates further improve when distributions are simultaneously smooth and supported on a set of low intrinsic dimension. It is easy to see that the empirical distribution does *not* benefit from assumed smoothness (see, e.g., Proposition 6 of Weed and Bach [65]). Whether an orthogonal series estimate benefits from low intrinsic dimension may depend on the basis used; the Fourier basis is not likely to benefit, but a wavelet basis, which is spatially localized, may. Nearest neighbor methods have also been shown to benefit from both smoothness and low intrinsic dimensionality, under $\mathcal{L}_\mu^2$ loss, and may therefore be promising [28].

The results in this paper should also be generalized to larger classes of spaces, such as inhomogeneous Besov spaces. Over these spaces, the classic work of Donoho et al. [15] suggests that simple linear density estimators such as the orthogonal series estimator studied in this paper cease to be minimax rate-optimal, but simple non-linear estimators such as wavelet thresholding estimators may continue to be (adaptively) minimax optimal.

The results of Yarotsky [67], on uniform approximation of smooth functions (over Sobolev spaces) by neural networks, we crucial to the result Theorem 7 bounding the error of perfectly optimized GANs. If these approximation-theoretic results can be generalized to other spaces (e.g., RKHSs), then our Theorem 1 can be used to derive performance bounds for perfectly optimized GANs over these spaces.

Finally, it has been widely observed that, in practice, optimization of GANs can be quite difficult [41, 34, 7]. This limits the practical implications of our performance bounds on GANs, which assumed perfect optimization (i.e., convergence to a generator-optimal equilibrium). Conversely, most work studying the optimization landscape of GANs is specific to the noiseless (i.e., "infinite sample size") case, whereas our lower bounds suggest that the sample complexity of training GANs may be substantial. Hence, it is important to generalize these statistical results to the case of imperfect optimization, and, conversely, to understand the effects of statistical noise on the optimization procedure.