[Reviews · NeurIPS 2018]

Reviewer 1



This paper derived under certain conditions the minimax convergence rate of nonparametric density estimator formed by truncated series estimator with terms of mapping to orthonormal basis under adversarial losses. The rate is approximately n^{-\frac{s+t}{2t+d}} under Sobolev space and a similar form under GAN. Under further assumptions on risk etc., the proposed minimax results for density estimation imply results related with implicit generative modeling. This is a nicely written paper with clear logic and meaningful theoretical result relating traditional nonparametric estimation and recent practice like GAN. A few detailed comments below: 1) Define notations like Line 49 \mathcal{L}^{p} when first appear, Line 75 \mathcal{Z}, … and Line 81 f^{(s)} briefly in this paper may help presentation. 2) Line 80 should be p\geq 1, Formula (3) latter part should be \hat{P}_{z}\phi_{z}=\frac{1}{n}… 3) Some “by default” assumptions like existence of sup{a_{z}^{-1}} can be understood, but would be nice if all assumptions were stated out together. 4) Line 82-84: Browsed the proofs but not too carefully, not sure which are all the places that the bounded condition was required and whether can be relaxed with a little more effort in maybe a future work (maybe just my biased preference/two cents)

Reviewer 2



Overview: This paper looks at nonparametric density estimation under certain classes of metrics, which the authors call "adversarial losses". To define adversarial losses, assume we have two probability distributions P and Q, and suppose X~P and Y~Q. Consider the supremum of |E[f(X)] - E[f(Y)]|, when f ranges over a class F_d of functions. This is the "adversarial loss with respect to class F_d", or simply the "F_d-metric", and generalizes several metrics, for instance the L_1 metric. Now, assume P is an unkown distribution belonging to some known class F_G, and we have n i.i.d. samples from P, and want to output an estimate another distribution Q in F_G so that the F_d-distance between P and Q is minimized. How small can we make this distance? Theorem 1 in this paper gives an upper bound for this question, and Theorem 3 gives a lower bound. Both bounds make smoothness assumptions about functions in F_d and F_G, and the bounds are tight for certain Sobolev spaces (Example 4). The upper bound is proved via an orthogonal series estimator. The motivation for this metric comes from generative adversarial networks (GANs), where the class F_d corresponds to functions that can be realized by a neural network with a certain architecture (the discriminator). In Theorem 7, the authors prove a certain performance guarantee for GANs (assuming perfect optimization), which improves the recent paper [25]. Finally, the paper compares the sample complexity of "explicit density estimation" as defined above, where the goal is to output a description of a distribution Q that is close to P, versus implicit density estimation, where the goal is to output a "generator function" that can generate i.i.d. samples from a distribution Q close to P. The main result here is Theorem 9, which states that, under certain assumptions, the two tasks have the same sample complexity up to constant factors. This result is the first of this spirit as far as I know. The paper is notationally quite heavy and I haven't understood all results completely. I have also not checked any of the proofs, which appear in the supplementary material. Pros: + given the current interest on GANs, the sample complexity result of this paper can be of significant interest. + the notion of density estimation with respect to adversarial losses is a novel notion introduced recently, and this paper makes significant progress towards understanding this problem. In some sense this paper connects the classical density estimation literature with the new developments on generative models. + the connection between implicit density estimation and explicit density estimation, made in this paper (Theorem 9), is novel. + the results are mathematically interesting and quite non-trivial. Cons: - there are lots of typos, and many of the terms have not been defined properly. Comments for the author: * Line 26: losses -> metrics * Line 45: linear projection -> mapping * Line 49-50: delete "(1-)" * Line 66: 3 -> Section 3 * Line 80: q -> p * Line 80: please define Lebesgue norm. Also what is mu? Is mu = lambda? Please clarify. * Line 81: please define the sth-order fractional derivative, rather than referring the reader to a book. It is a central definition in your paper, which must be standalone. * Line 85: is well-defined "for any P." * Line 85: define real-valued net. * Equation (4): hat{P} -> hat{P}_Z * Line 165: delete the second "that" * Line 172: define "parametric convergence" * Line 198: zeta^d -> (2zeta+1)^d * In Theorem 7, what is d ? * In Section 7, define the GAN setup (don't assume the reader knows what is the generator and what is the discriminator) * Line 255: on -> of * Line 265: a latent -> the latent * Line 266: missing closed brackets * Line 280: delete "below" * Line 296: no -> not * Line 318 and 319: estimators -> samplers * Section 8 is rather different from the rest of the paper. I would suggest moving it to the appendix, and using the opened up space to define all the used terms properly and elaborate more on the definitions and explain the techniques and intuition behind proofs. * In Lines 49-51 you mention many of the common metrics can be written in terms of F_d-metrics. Elaborate please. * For Theorem 1, do you need to make any assumptions about the orthogonal basis B? == After reading other reviews and authors' responses: thanks for the answers. Definition of f^{(s)}: please mention the page number of the book on which this is defined to ease the reader. Line 26: currently the sentence reads "loss is measured not with L^p distances but rather with weaker losses" which is awkward. Lines 49--51: please mention mu is Lebesgue measure.

Reviewer 3



This paper is about non parametric density estimation. The quality of an estimator is evaluated using adversarial losses, for example maximum mean discrepancy, Wasserstein distance, total variation. The estimator is built using the first coefficients of a given orthogonal basis, specifically all the elements of the basis with indices lower that a threshold. The authors provide an upper bound for this estimator when the classes of functions are smooth that improves previous results. They also give a lower bound and show on several examples that the rates match. The estimator depend on the threshold and in order to get adaptive result, the author build an estimator of the threshold and prove that they can still obtain the same rates. Finally, they apply their results in the case of GANs and the sampling problem. The paper is well written and easy to follow. Looking at density estimation through the adversarial losses is quite new and the results of the paper are the ones we are looking for. This work can really help to understand the nature of GANs and VAEs and could be helpful for practitioner that use it. Even though I did not check line by line the proofs, the ideas are quite classical and I am confident that the results can be reproduced.

Reviewer 4



This paper studies minimax rates for nonparametric density estimation using integral probability metrics, when densities can be expressed in a countable basis of a Sobolev-type ellipsoid. Pros: important problem, the IPMs bring an interesting application to GANs. I have appreciated the discussion after Theorem 9. Cons: Paper that looks simultaneously mathy and drafty at the same time (see comments). The papers key contributions are in fact looser than expected from reading the paper: gap between Th. 1 and 3 (see below), constants that are not constants in Example 4 (see below), so-called tightening of Liang's results in Theorem 7 that is in fact far from obvious (see below). Fixes to consider (L607-L608, see below). Details: Theorem 1 vs Theorem 3 : unless significant restrictions apply, there is a huge gap between the two Theorems due to the fact that the upperbound in Theorm 1 depends on 1/(inf a_z) * 1/(inf b_z) while the lowerbound in Theorem 3 depends on 1/(sup a_z) * 1/(sup b_z) Theorem 1 : I do not agree with L156 -- we cannot say that the second term decreases with Z at a rate depending on the complexity of F_G, because it involves a tricky dependence between L_G and b. Example 4: c and C are not constants: they depend on d. Furthermore, \zeta is not "some" real >0. It in fact depends on n (L616 - L617 Appendix). This makes very hard to read (6) and it is certainly not as easy at it looks from just (6). Theorem 7: in L225, the authors say that the upperbound of Liang is loose, so the authors' Theorem is better. But loose in what sense ? There is absolutely no insights into how we should consider Theorem 3.1 in Liang "loose" with respect to the paper. Furthermore, there is no proof of the Theorem, which makes even harder the task of comparing the results (I would have loved to see the expression of C in L237 that the authors come with). Instead of comments like L228-L229, why not expanding the proof in the Appendix so we can have a concrete idea ? Appendix: * L568-L569: there lacks the |.|^2 in tilde P_z in the first ineq. * L596-L597: notation || * L607-L608: I do not follow the third to last inequality, which would be a natural consequence of assuming p_\tau(x) \geq 1/2 -- that you do not assume -- but otherwise seems hard to obtain. * (16): missing =0. I do not like the drafty-look of the Apppendix, in particular the TODO list of things to do, change, amend, etc in red. This gives the impression that the authors went very fast into proving some things -- perhaps too fast sometimes: it is not acceptable to have L662-L663, since you have no idea of what c looks like -- it is certainly not a constant. Typoes: L71: remove "to denote [...] at most n". This statement is useless (and in fact, wrong as stated).